# OptMATH: A Scalable Bidirectional Data Synthesis Framework for Optimization Modeling

Hongliang Lu [* 1]  Zhonglin Xie [* 2]  Yaoyu Wu [1]  Can Ren [3]  Yuxuan Chen [3]  Zaiwen Wen [2]

## Abstract

Despite the rapid development of large language models (LLMs), a fundamental challenge persists: the lack of high-quality optimization modeling datasets hampers LLMs' robust modeling of practical optimization problems from natural language descriptions (NL). This data scarcity also contributes to the generalization difficulties experienced by learning-based methods. To address these challenges, we propose a scalable framework for synthesizing a high-quality dataset, named OptMATH. Starting from curated seed data with mathematical formulations (MF), this framework automatically generates problem data (PD) with controllable complexity. Then, a back-translation step is employed to obtain NL. To verify the correspondence between the NL and the PD, a forward modeling step followed by rejection sampling is used. The accepted pairs constitute the training part of OptMATH. Then a collection of rejected pairs is identified and further filtered. This collection serves as a new benchmark for optimization modeling, containing difficult instances whose lengths are much longer than these of NL4OPT and MAMO. Through extensive experiments, we demonstrate that models of various sizes (0.5B-32B parameters) trained on OptMATH achieve superior results on multiple modeling benchmarks, thereby validating the effectiveness and scalability of our approach. The OptMATH dataset and related resources are available at https://github.com/optsuite/OptMATH.

---

[*]Equal contribution, alphabetical order [1]College of Engineering, Peking University [2]Beijing International Center for Mathematical Research, Peking University [3]School of Mathematics Science, Peking University. Correspondence to: Zaiwen Wen <wenzw@pku.edu.cn>.

*Proceedings of the $42^{nd}$ International Conference on Machine Learning*, Vancouver, Canada. PMLR 267, 2025. Copyright 2025 by the author(s).

## 1. Introduction

Automatic translation of natural language descriptions of optimization problems into solver-ready formats is a critical step in democratizing access to optimization techniques. This capability would enable individuals without expertise in optimization to leverage the power of optimization for solving real-world problems across various domains, including logistics (Ghiani et al., 2022), finance (Consigli, 2019), and engineering (Rao, 2019). Known as optimization modeling, this task has long been challenging due to the inherent ambiguity of natural language and the need for a deep understanding of optimization modeling principles. The manual process of formulating an optimization problem typically involves iterative refinement and demands significant mastery of the relevant techniques, making it time-consuming and inaccessible to many practitioners.

Recent advances in Large Language Models (LLMs), such as ChatGPT (Brown et al., 2020b), GPT-4 (OpenAI et al., 2023), and OpenAI's o1 (OpenAI, 2024), have demonstrated remarkable capabilities in understanding natural language and performing complex reasoning tasks. Notably, the introduction of o1 has significantly enhanced the performance of LLMs in mathematical reasoning, achieving state-of-the-art results on challenging datasets including AIME (2024), GPQA, and CodeForces. However, optimization modeling presents unique challenges. Unlike grade-school mathematics, where problems typically have a single correct solution, optimization problems can often be approached using multiple valid models, making the task semi-open-ended. Furthermore, optimization frequently relies on a vast body of empirical knowledge that is less formally structured than purely mathematical concepts. As a result, research has shown that directly applying LLMs to optimization modeling tasks yields suboptimal outcomes (Ramamonjison et al., 2021).

**LLMs for Optimization Modeling.** To address this issue, recent efforts have explored various strategies. Research on leveraging LLMs for optimization modeling typically follows two main approaches. The first uses prompt engineering techniques to guide LLMs in generating or refining optimization models, without modifying the underlying model parameters. Examples include the NL4Opt compe-

tition (Ramamonjison et al., 2021), which aims to extract optimization formulations from natural language descriptions, and OptiMUS (AhmadiTeshnizi et al., 2023), which proposes an agent-based prompt engineering method. More recently, Autoformulation (Astorga et al., 2024) combines Monte Carlo tree search with LLMs to address optimization modeling. While these methods rely heavily on the base capabilities of general-purpose LLMs, they do not yield fundamental advancements beyond refined prompting.

The second approach centers on fine-tuning, wherein model parameters are adapted using specially curated or synthesized optimization datasets. ORLM (Tang et al., 2024) introduces OR-Instruct, a data augmentation framework for optimization problems, and demonstrates performance gains via Supervised Fine-Tuning on a foundation model. LLMOPT (Jiang et al., 2024) likewise applies a multi-instruction fine-tuning strategy and self-correction mechanisms. Similarly, OptLLM (Zhang et al., 2024) presents a framework that fine-tuned a Qwen model using a large-scale self-developed optimization dataset, reporting an accuracy boost over prompt-based models. However, many of these methods still tend to generate synthetic training data that can be limited in quantity, or may lack consistent quality and sufficient complexity. Consequently, their ability to generalize to more sophisticated optimization tasks can be constrained.

**LLM-Based Data Synthesis for Optimization.** Data synthesis methods have become essential for addressing data scarcity and enhancing the performance of LLMs. According to (Wang et al., 2024), these methods can be broadly categorized into two main approaches: data augmentation and data synthesis. OR-Instruct (Tang et al., 2024; Wang et al., 2022) exemplifies a data augmentation approach, following the self-instruct framework to expand existing datasets. In (Yang et al., 2025), a framework is proposed for the reverse synthesis of optimization problem data pairs. However, this approach initiates reverse synthesis from formatted demonstrations, which inherently limits its scalability and does not ensure the correspondence between the generated code and the natural language descriptions. MILP-Evolve (Li et al., 2024) introduces an evolutionary framework designed to generate diverse mixed-integer linear programming (MILP) problems using LLMs. While MILP-Evolve and other synthesis efforts represent significant steps forward in synthetic data generation, the critical challenge of translating natural language descriptions into robust mathematical optimization models often remains a key area for advancement, which our OptMATH framework directly addresses through its scalable, bidirectionally validated approach.

**Instance Generation for Optimization.** Recent research in MILP instance generation has evolved along two primary axes: learning-based structural synthesis and rule-based distribution expansion. The learning-based paradigm ad-

dresses data scarcity by developing generative models that preserve instance hardness and constraints. Examples include the bipartite graph variational autoencoder framework proposed by (Geng et al., 2023), the block decomposition operators for constraint matrices introduced in (Liu et al., 2024b), the duality-driven feasibility guarantees established by (Wang et al., 2023), and the adaptive constraint modification mechanisms developed in (Guo et al., 2024). The rule-based approach leverages instance space analysis techniques (Alipour et al., 2022; Strassl & Musliu, 2022; Alipour & Smith-Miles, 2023) to guide the generation of more diverse instance distributions (Smith-Miles & Bowly, 2015; Bowly, 2019). Alternatively, it may rely on manually selected features to control the characteristics of the generated instances (Bowly et al., 2020).

**Our Contributions.** We propose a scalable bidirectional synthesis framework that addresses the critical challenge of data scarcity in optimization modeling through *triplet-aligned (NL, MF, PD) data generation* and rigorous validation. Our framework uniquely integrates a closed-loop workflow with *optimal value matching*, ensuring semantic equivalence between NL, MF, and PD. This approach demonstrates exceptional scalability. The framework's domain adaptability is evidenced by coverage of 10+ real-world applications (e.g., logistics, energy, finance) through 53 seed generators, with manual analysis confirming 99.6% equivalence accuracy across all triplets.

We introduce *OptMATH-Train*, a large scale verified optimization modeling dataset containing rigorously validated (NL, MF, PD) triplets. Each triplet undergoes three-stage quality control: mathematical consistency checks (MF-to-PD compilation), semantic fidelity validation (PD-to-NL backtranslation), and solution equivalence verification through solver-based rejection sampling. From rejected instances, we curate *OptMATH-Bench*, a challenging benchmark comprising "hard instances" characterized by extended natural language contexts ($2.9\times$ longer than MAMO EasyLP) and complex constraints. We further span it using various problems including LP, MILP, IP, NLP, SOCP. This benchmark provides the standardized evaluation for long-context optimization modeling.

Finally, extensive experiments demonstrate the efficacy of our framework. Models trained on the OptMATH-Train dataset achieve state-of-the-art performance on multiple established modeling benchmarks, including NL4OPT and MAMO. These results definitively validate the effectiveness and scalability of our framework for generating high-quality optimization modeling datasets.

## 2. Backgrounds & Overview

In mathematical optimization theory, a canonical optimization problem can be formulated as:

$$\begin{aligned} \min_{\mathbf{x}} \quad & g(\mathbf{x}), \\ \text{subject to} \quad & c_i(\mathbf{x}) = 0, \quad i \in \mathcal{E}, \\ & c_i(\mathbf{x}) \geq 0, \quad i \in \mathcal{I}, \end{aligned} \quad (1)$$

where $\mathbf{x} \in \mathbb{R}^n$ denotes the decision vector. The objective function $g : \mathbb{R}^n \to \mathbb{R}$ assigns a scalar value to each candidate solution, which we seek to minimize. The constraint functions $c_i : \mathbb{R}^n \to \mathbb{R}$ define the feasible region through equality constraints indexed by $\mathcal{E}$ and inequality constraints indexed by $\mathcal{I}$.

To formalize the optimization modeling problem, we define several key concepts and illustrate them using a concrete example in Appendix D.1. For a specific optimization problem, we define NL as the *natural language description*, which corresponds to the "Input-Natural Language Description" in the example. We represent the LP/MPS file, the concrete mathematical expression (corresponding to the "Output-Instance Formulation"), and any other solver-ready formats, such as executable Python code with Gurobi shown in Appendix D.1, as *problem data* (PD). A common characteristic of these representations is that they allow us to obtain the optimal value of the problem by invoking a solver based on the PD. *Mathematical formulation* (MF) refers to formulation where concrete numbers are not yet specified, corresponding to the "Output-General Formulation" in the example. We emphasize that, in subsequent sections, we may use different forms of PD. However, these forms essentially carry the same information about the problem and can be inferred from the context without ambiguity. We use different forms of PD to facilitate their integration into the workflow and to enhance clarity in various contexts.

Modern solvers such as Gurobi and Mosek (Gurobi Optimization, LLC, 2024; MOSEK ApS, 2025) can efficiently solve problems stored with PDs using algorithms like interior-point methods (Karmarkar, 1984). However, practical challenges remain. In real-world applications, one of the main difficulties lies in converting informal NLs of problems into precise MFs. Moreover, extracting the PDs from NLs poses an additional significant challenge. Traditionally, this process has required deep optimization expertise (Boyd, 2004), but recent advances in LLMs offer promising opportunities to automate this transformation.

Let $\mathcal{A}_\theta$ represent an LLM parameterized by $\theta$. The formulation for increasing the modeling capability of the LLM can be expressed as:

$$\max_{\theta} \quad \mathbb{E}_{(\text{NL},\text{MF},\text{PD}) \sim \mathcal{D}}[Q_{(\text{NL},\text{MF},\text{PD})}(\text{MF}', \text{PD}')], \quad (2)$$

$$\text{s.t.} \quad (\text{MF}', \text{PD}') = \mathcal{A}_\theta(\texttt{prompt}_{\text{M}}(\text{NL})), \quad (3)$$

where $\texttt{prompt}_{\text{M}}$ is a modeling prompt template mapping NL to $\text{MF}'$ and $\text{PD}'$. The quality metric $Q$ evaluates the generated $(\text{MF}', \text{PD}')$ pairs based on the verified $(\text{NL}, \text{MF}, \text{PD})$ triplet. Constraint (3) formalizes the automated formulation process (Autoformulation), as depicted in the example in Appendix D.1. Optimizing $\theta$ relies on having a large corpus of high-quality triplets $(\text{NL}, \text{MF}, \text{PD})$. To address this requirement, we propose a systematic framework for generating synthetic training data that maintains mathematical rigor.

**An Overview of Our Pipeline.** The pipeline of our framework is presented in Figure 1. In the reverse data generation phase, we collect optimization problems from two sources: (1) LP/MPS files from challenging benchmarks such as MIPLIB 2017 (Gleixner et al., 2021) and netlib (Netlib, 1990; Gay, 1985), and (2) over 50 expert-curated seed problem generators covering diverse optimization scenarios. Through our carefully designed backtranslation pipeline, we leverage both the LP files and their MFs to generate high-quality NLs of optimization problems. Notably, using an LLM-based feedback workflow with evaluation, our collected generators can produce tremendous PDs with controllable varying difficulty levels. This enables us to effectively address the data scarcity challenge in training learning-based optimization methods.

In the forward modeling and evaluation process, we utilize our trained AutoFormulator to translate the generated NLs back into PDs. Specifically, in this phase, all PDs are represented as solver code, which can then be exported as LP files. We then implement a rigorous rejection sampling strategy, where only instances whose optimal objective values match between the original and generated LP files are retained. This equivalence-based filtering mechanism ensures the high quality of our OptMATH-Train dataset by guaranteeing the semantic consistency of each instance.

Building upon the high-quality instances obtained through rejection sampling, we employ various data augmentation strategies to further enhance the diversity and coverage of our dataset. This enriched collection of training pairs is then utilized to fine-tune a foundation model, leading to AutoFormulator, a specialized model specifically designed for automated mathematical optimization modeling.

## 3. Feedback-Driven PD Generation

The mature development of the optimization community has provided us with access to many high-quality optimization PDs. These PDs are typically stored in standardized formats like MPS or LP files. To effectively leverage these resources, we began by curating over 50 seed problem classes sourced from a variety of optimization journals and websites (see Appendix A.2 for details). For each $i$-th problem class, we

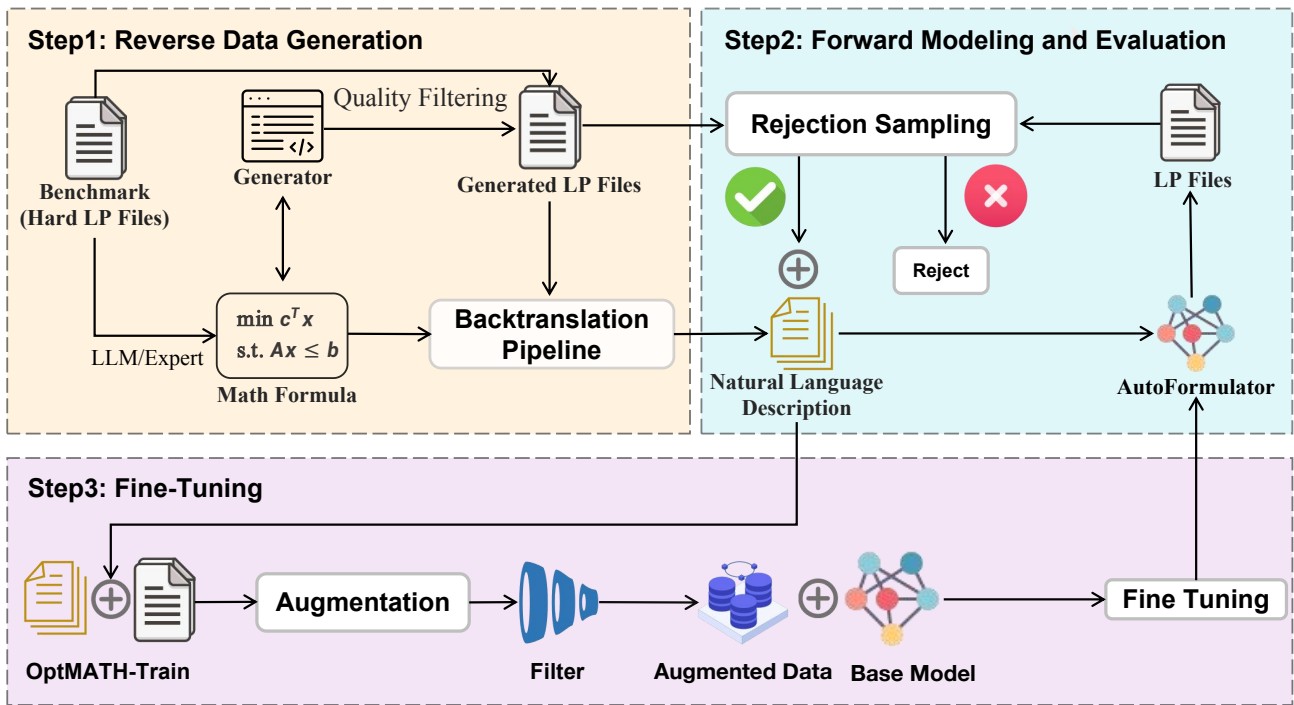

*Figure 1.* An overview of our scalable, bidirectional data synthesizing pipeline.

developed a corresponding instance generator $G_i$. This generator takes a problem-specific configuration as input and outputs a probability distribution over PDs. The distribution is designed to produce PDs with varying scales and complexities, which are controllable through adjustable configurations. Before delving into the controlled generation process for the PDs, we first explain how the complexity of PDs can be measured.

**Measuring the Modeling Complexity.** The complexity of formulating and solving a MIP problem depends on modeling choices such as the types of variables, the forms of constraints, and auxiliary modeling techniques employed. We introduce a scoring function $S$ defined as:

$$
\begin{aligned}
S(\text{PD}) = & \; \alpha_{\text{bin}} N_{\text{bin}} + \alpha_{\text{int}} N_{\text{int}} + \alpha_{\text{cont}} N_{\text{cont}} \\
& + \beta_{\text{lin}} N_{\text{lin}} + \beta_{\text{indic}} N_{\text{indic}} + \beta_{\text{quad}} N_{\text{quad}} \\
& + \beta_{\text{gen}} N_{\text{gen}} + \gamma_{\text{BigM}} f_{\text{BigM}} + \delta_{\text{expr}} \overline{L_{\text{expr}}},
\end{aligned}
\tag{4}
$$

where $N_{\text{bin}}, N_{\text{int}}, N_{\text{cont}}$ are the number of binary, integer, and continuous variables, respectively. Similarly, $N_{\text{lin}}, N_{\text{indic}}, N_{\text{quad}}, N_{\text{gen}}$ represent the number of linear, indicator, quadratic, and general nonlinear constraints. The term $f_{\text{BigM}}$ is a factor reflecting the frequency of Big-M formulations, and $\overline{L_{\text{expr}}}$ is the average number of terms per constraint and the objective function, which captures the structural information of the expressions. Lastly, the weights $\alpha_{\cdot}, \beta_{\cdot}, \gamma_{\text{BigM}}, \delta_{\text{expr}}$ are tunable parameters reflecting the con-

tribution of each component to the overall complexity. To illustrate it, we provide an example in Appendix B.1 that considers a MIP problem incorporating multiple constraint types to optimize production costs for two products under resource constraints.

**Selecting Parameters to Control the Complexity.** We now present the workflow in Algorithm 1 for selecting parameter configurations for instance generator $G_i$ to generate PDs fitting the complexity, feasibility, and solving time requirements. The prompt templates are illustrated in Appendix E.4. As formalized in Algorithm 1, the process begins by specifying target bounds. Then a template $\text{prompt}_{\text{IC}}$ for initializing the configuration are incorporated. After obtaining the configuration, we generate $N$ PDs using it. We then evaluate the generated PDs through the complexity score, solving time, and feasibility satisfactory. Then, a feedback $\text{prompt}_{\text{RC}}$ is created based on the statistics of these metrics over the $N$ generated PDs. The LLM iteratively adjusts parameters based on feedback from solved instances, ultimately converging to a configuration that satisfy predefined criteria. This ensures generated PDs remain both expressive and tractable by adhering to runtime thresholds.

## 4. The Data Synthesis Framework

This section presents our bidirectional scalable data synthesis framework. In this section, all of the PDs are in solver

---

**Algorithm 1** Feedback-Driven Problem Data Generation

**Require:** Target complexity range $[S_{\min}, S_{\max}]$, time limits $[T_{\min}, T_{\max}]$, instance generator $G$, feasibility threshold $\mathcal{F}_{\text{target}}$, max iterations $T$

**Ensure:** Configuration $\Theta$ such that for $\text{PD}_i \sim G(\Theta)$:
$S(\text{PD}_i) \in [S_{\min}, S_{\max}]$ (complexity), $\tau_i \leq T_{\max}$ (solving time), $\Pr(f_i = \texttt{feasible}) \geq \mathcal{F}_{\text{target}}$

1: Initialize parameters via LLM:
   $\Theta_0 \leftarrow \mathcal{L}(\texttt{prompt}_{\text{IC}}(S_{\min}, S_{\max}, T_{\min}, T_{\max}))$
2: **for** $t = 1$ **to** $T$ **do**
3:   Generate $N$ PDs: $\{\text{PD}_i\}_{i=1}^N \leftarrow G(\Theta_{t-1})$
4:   Compute metrics: $S(\text{PD}_i)$ (Eq. 4), $\tau_i$ (solving time), $f_i$ (feasibility)
5:   Aggregate statistics: $\bar{S}_t = \frac{1}{N} \sum S(\text{PD}_i)$, $\bar{\tau}_t = \frac{1}{N} \sum \tau_i$, $\mathcal{F}_t = \frac{1}{N} \sum \mathbb{I}(f_i = \texttt{feasible})$
6:   **if** $\bar{S}_t \in [S_{\min}, S_{\max}]$ **and** $\bar{\tau}_t \leq T_{\max}$ **and** $\mathcal{F}_t \geq \mathcal{F}_{\text{target}}$ **then**
7:     **return** $\Theta_{t-1}$
8:   **else**
9:     Refine parameters via feedback:
       $\Theta_t \leftarrow \mathcal{L}(\texttt{prompt}_{\text{RC}}(\bar{S}_t, \bar{\tau}_t, \mathcal{F}_t; \Theta_{t-1}))$
10:  **end if**
11: **end for**
12: **return** $\emptyset$ (no valid $\Theta$ found)

---

**Algorithm 2** Bidirectional Data Synthesis Algorithm

**Require:** Instance pair $(\text{MF}_i, \text{PD}_{i,j})$, Max Iteration $T$

**Ensure:** $(\text{NL}_{i,j}, \text{MF}'_{i,j}, \text{PD}'_{i,j}, \text{OV}_{i,j})$

1: Initial generation: $\text{NL} \leftarrow \mathcal{L}(\texttt{prompt}_{\text{I}}(\text{MF}_i, \text{PD}_{i,j}))$
2: Initialize: $\text{SC} = \text{SR} = \text{Null}$
3: **for** $k = 1, \ldots, T - 1$ **do**
4:   Self-Criticize:
     $\text{SC} \leftarrow \mathcal{L}(\texttt{prompt}_{\text{C}}(\text{MF}_i, \text{PD}_{i,j}, \text{NL}))$
5:   Self-Refine:
     $\text{SR} \leftarrow \mathcal{L}(\texttt{prompt}_{\text{R}}(\text{MF}_i, \text{PD}_{i,j}, \text{NL}, \text{SC}, \text{SR}))$
6:   **if** SR is good enough **then**
7:     **break**
8:   **end if**
9: **end for**
10: $\text{NL}_{i,j} \leftarrow \text{SR}$
11: AutoFormulation:
    $(\text{MF}'_{i,j}, \text{PD}'_{i,j}) \leftarrow \mathcal{A}_\theta(\texttt{prompt}_{\text{M}}(\text{NL}_{i,j}))$
12: $\text{OV}_{i,j} \leftarrow$ Solve $\text{PD}_{i,j}$ by Gurobi
13: $\text{OV}'_{i,j} \leftarrow$ Solve $\text{PD}'_{i,j}$ by Gurobi
14: **if** $\text{OV}_{i,j} = \text{OV}'_{i,j}$ **then**
15:   **return** $(\text{NL}_{i,j}, \text{MF}'_{i,j}, \text{PD}'_{i,j}, \text{OV}_{i,j})$
16: **else**
17:   **return** Null
18: **end if**

---

code form (see subsection 4.2 for more details). Let $\mathcal{L}$ represents the LLM employed on the reverse data generation phase, and $\mathcal{A}_\theta$ for our fine-tuned AutoFormulator with weights $\theta$. We define $\texttt{prompt}_{\text{I}}$, $\texttt{prompt}_{\text{C}}$, $\texttt{prompt}_{\text{R}}$ as the prompt templates that accept certain inputs for the initial generation, self-critism, and self-refinement stages. The algorithm is formalized in Algorithm 2, with an illustrative example of the backtranslation process shown in Figure 13.

The final OptMATH dataset $\mathcal{D}$ is constructed by collecting all valid quadruples $(\text{NL}_{i,j}, \text{MF}'_{i,j}, \text{PD}'_{i,j}, \text{OV}_{i,j})$ that pass the validation process, where $\text{MF}'_{i,j}$ and $\text{PD}'_{i,j}$ represent the generated mathematical formulation and problem data using $\mathcal{A}_\theta$, $\text{OV}_{i,j}$ is the optimal value obtained by solving the problem specified by $\text{PD}_{i,j}$. By leveraging our instance generators and the iterative refinement process, this algorithm enables scalable generation of high-quality data pairs. The mathematical equivalence between the generated formulations and the original instances is rigorously validated through rejection sampling, ensuring the reliability of our dataset. A comprehensive discussion of our quality control and rejection sampling can be found in Section 4.3.

### 4.1. Backtranslation Pipeline

To generate high-quality NLs of optimization problems at scale, we leverage a specific LLM as the foundation of our pipeline. Recent research has demonstrated that complex tasks often benefit from iterative refinement approaches rather than direct generation (Madaan et al., 2024). This observation aligns with human problem-solving processes in mathematics, which typically requires multiple attempts and refinements. Building upon this insight, we design a three-phase backtranslation pipeline that systematically improves the quality of generated descriptions through iterative refinement. All prompt templates used in this pipeline can be found in E.1.

**Initial Generation.** Given the mathematical formulation $\text{MF}_i$ and the corresponding problem data $\text{PD}_{i,j}$ of a problem $j$ in $i$-class, the LLM generates an initial natural language description NL using the prompt template $\texttt{prompt}_{\text{I}}$. This stage requires the model to comprehend both the mathematical semantics and the instance parameters to produce a preliminary human-readable description.

**Self-Criticism.** Using prompt template $\texttt{prompt}_{\text{C}}$, the LLM evaluates the current description by examining the mathematical equivalence with $\text{MF}_i$, completeness of the constraints and objective functions, clarity and comprehensibility, and consistency of the parameters with $\text{PD}_{i,j}$. The criticism SC in iteration $k$ incorporates feedback from all previous iterations to guide improvements.

**Self-Refinement.** Based on the criticism, the model generates refined descriptions SR with the prompt template $\texttt{prompt}_{\text{R}}$. The refinement process focuses on improving

the mathematical accuracy, completeness of the constraints, and clarity of the descriptions.

This process iterates for $T$ rounds until a satisfactory description $\mathrm{NL}_{i,j}$ is obtained, with each iteration potentially improving the quality of the generated description. Based on our empirical analysis (see Appendix C.2), we set $T = 1$ in the final implementation.

### 4.2. Forward modeling

Building upon the NLs generated in subsection 4.1, we leverage AutoFormulator to transform them back into MFs and PDs in solver code form, enabling rejection sampling for quality validation. Given a NL as input, AutoFormulator produces two key outputs: a MF and corresponding PD in solver code form. While previous works (Tang et al., 2024; Jiang et al., 2024) adopted fixed output formats, our approach is not constrained to any particular format, as our primary goal is to obtain correct solver code, with the formulation serving as an intermediate reasoning step. To facilitate genuine mathematical modeling capabilities rather than superficial format mapping, we design diverse Chain-of-Thought (CoT) prompting strategies (Wei et al., 2022). This approach generates multiple valid reasoning paths and formulation variants for the same problem, enriching our training data with diverse modeling perspectives and enhancing the model's mathematical reasoning capabilities. Detailed implementation of these CoT strategies is described in Appendix D.1.

### 4.3. Rejection Sampling

To ensure the quality and mathematical soundness of our generated optimization problem descriptions, we employ a rejection sampling strategy (Yuan et al., 2023; Liu et al., 2024c) to filter and select high-quality samples from the generated candidates.

As illustrated in Algorithm 2, our rejection sampling mechanism relies on solution-based comparison to validate the generated samples. Specifically, for each generated natural language description $\mathrm{NL}_{i,j}$, we use AutoFormulator to transform it into a mathematical formulation $\mathrm{MF}'_{i,j}$ and solver code $\mathrm{PD}'_{i,j}$, obtaining solution $\mathrm{OV}'_{i,j}$. This solution is then compared with $\mathrm{OV}_{i,j}$, obtained by directly solving the original instance $\mathrm{PD}_{i,j}$. A sample is accepted into our dataset $\mathcal{D}$ as a validated quadruple $(\mathrm{NL}_{i,j}, \mathrm{MF}'_{i,j}, \mathrm{PD}'_{i,j}, \mathrm{OV}_{i,j})$ if and only if $\mathrm{OV}_{i,j} = \mathrm{OV}'_{i,j}$.

While this solution-based validation approach may not guarantee perfect equivalence (as problems with identical optimal values may represent different optimization problems), our analysis using an LLM-committee (Zhao et al., 2024) and manual inspection of randomly sampled instances (1% of the total dataset) reveals a remarkable 99.6% accuracy

rate. We acknowledge that determining the exact equivalence between two mathematical formulations remains an open research question worthy of further investigation. Nevertheless, our current approach provides a practical and highly effective mechanism for ensuring dataset quality.

## 5. Fine-Tuning

### 5.1. Data Augmentation

To improve the diversity of our dataset, we use data augmentation to augment the training data. This method generates more non-standard problems compared to a data generator, enhancing the model's generalization performance. We create rules for problem rewriting, semantic substitution, constraint expansion, and numerical augmentation. For each instance, a randomly selected rule is used to prompt the LLMs to generate the corresponding augmented data. The detailed augmentation rules and prompt templates can be found in Appendix E.7.

For quality control, we employ a specific LLM $\mathcal{L}$ to sample each augmented description twice independently, followed by the rejection sampling strategy described in Section 4.3. This process yields approximately 10 qualified augmented datasets for each problem, and this method was applied to augment 50 thousand instances to complement our original dataset.

### 5.2. Training the AutoFormulator

We adopt a supervised fine-tuning (SFT) approach to enhance the AutoFormulator's modeling capabilities. Specifically, we employ the LoRA algorithm (Hu et al., 2021) for efficient parameter-efficient fine-tuning, which significantly reduces memory requirements while maintaining model performance by updating only a small set of adapter parameters. Using the OptMATH-Train dataset $\mathcal{D}_{\mathrm{SFT}} = \{(\mathrm{NL}_i, \mathrm{MF}_i, \mathrm{PD}_i)\}_{i=1}^{N_{\mathrm{Train}}}$, we train the model to generate both mathematical formulations and solver code given problem descriptions. For each training sample, the input consists of the problem description $\mathrm{NL}_i$, while the target output is the concatenation of the formulation and solver code: $y_i = [\mathrm{MF}_i; \mathrm{PD}_i]$, where $[;]$ denotes sequence concatenation. The training objective follows the standard sequence-to-sequence loss:

$$\mathcal{L}_{\mathrm{SFT}}(\theta) = -\mathbb{E}_{(p,y)\sim\mathcal{D}_{\mathrm{SFT}}^A}\left[\sum_{t=1}^{|y|} \log P_\theta(y_t|y_{<t}, p)\right] \quad (5)$$

where $y_t$ represents the token at position $t$ in the target sequence, and $y_{<t}$ denotes all preceding tokens. This approach allows the model to learn the mapping from natural language problem descriptions to both mathematical formulations and solver code within a unified sequence-to-

sequence framework.

# 6. Experiments

## 6.1. Statistics of the OptMATH Dataset

First, using our generators, we generated a quality-filtered dataset containing over 600,000 LP files, which span 53 distinct problem types and are distributed across five hardness levels. For more details on the seed data class, please refer to Appendix A.2. To ensure computational feasibility, we impose a solving time threshold and employ a feedback pipeline that leverages an LLM to regulate both the complexity and feasibility of the generated instances. Further details on this process are provided in Appendix B. The distribution of file lengths across these LP files is visualized in Figure 2. As shown, the lengths range widely from 1,000 to 25,000 characters, capturing a rich variety of problem complexities. The proportions of different lengths are well-balanced, with a concentration on medium difficulty levels (which are already quite challenging compared to other benchmarks) and a gradual decline as the problems become harder. Additionally, the distribution confirms the effectiveness of our complexity control mechanism.

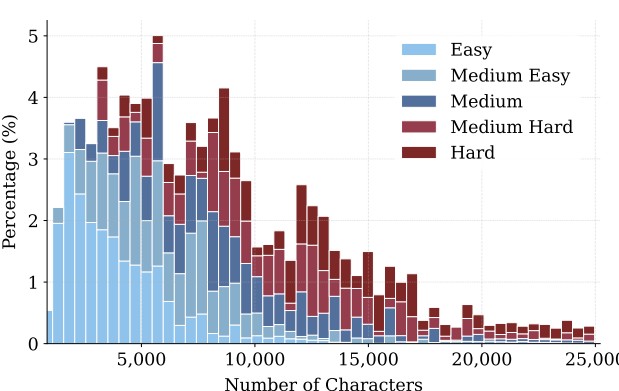

*Figure 2.* Distribution of LP file lengths.

We further conducted a comparative analysis of problem lengths between OptMATH and other benchmark datasets, with their average lengths shown in Figure 3. The analysis reveals that OptMATH presents significantly more complex problem descriptions compared to existing benchmarks. This increased complexity, manifested through longer problem descriptions, poses greater challenges for LLMs, as longer descriptions typically demand enhanced comprehension and reasoning capabilities.

As shown in Figure 4, OptMATH-Bench has selected a number of representative mathematical optimization problems covering a wide range of application scenarios, including LP, MILP, IP, NLP, SOCP and other optimization problems. For details, please refer to Appendix A.1.

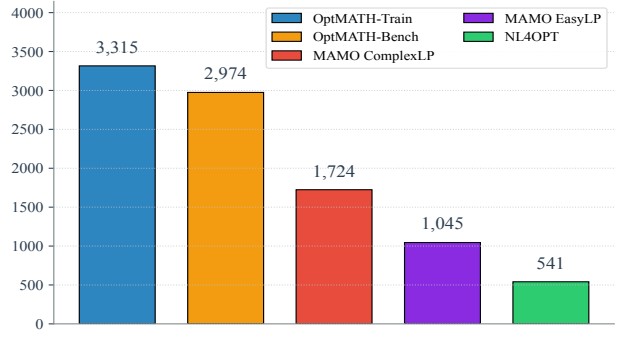

*Figure 3.* Question length analysis

| Type | NL4OPT | MAMO EasyLP | MAMO ComplexLP | OptMATH-Bench |
|------|--------|-------------|----------------|---------------|
| IP | 26.94% | 58.74% | 15.64% | 11.92% |
| LP | 61.63% | 0.61% | 62.08% | 8.81% |
| MILP | 11.43% | 40.65% | 22.28% | 67.87% |
| NLP | 0.00% | 0.00% | 0.00% | 5.18% |
| SOCP | 0.00% | 0.00% | 0.00% | 6.22% |

*Figure 4.* The proportion of problems in different datasets.

To visualize the distribution of different benchmarks and OptMATH dataset, we project their high-dimensional embeddings onto a 2D space using t-SNE(van der Maaten & Hinton, 2008). As shown in Figure 5, the instances from different sources form distinct clusters, suggesting that OptMATH effectively captures the diversity of different problem families. It can be observed that OptMATH surrounds the area of other benchmarks. This explains the improvement on various benchmarks obtained by training on OptMATH-Train.

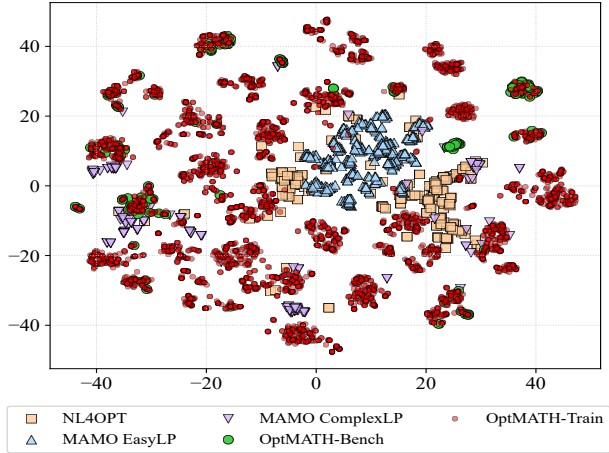

*Figure 5.* Visualization of OptMATH and other benchmarks.

## 6.2. Autoformulation

**Evaluation Benchmarks and Metrics.** We evaluate our fine-tuned model on five benchmarks: NL4OPT(Ramamonjison et al., 2021), MAMO(Huang et al., 2024), IndustryOR(Tang et al., 2024), OptiBench(Yang et al., 2025) and our newly constructed OptMATH-Bench.

*Table 1.* Performance Comparison of Models on Different Benchmarks

| Types | Models | NL4OPT | MAMO EasyLP | MAMO ComplexLP | OptMATH Bench | IndustryOR | OptiBench | Macro AVG |
|---|---|---|---|---|---|---|---|---|
| **Baseline** | Llama3.1-8B(pass@1) | 0.0% | 0.2% | 0.0% | 0.0% | 0.0% | 0.0% | 0.1% |
| | Qwen2.5-7B(pass@1) | 86.9% | 83.6% | 21.8% | 1.6% | 10.0% | 36.2% | 40.0% |
| | GPT-3.5-turbo(pass@1) | 78.0% | 79.3% | 33.2% | 15.0% | 21.0% | 47.4% | 51.4% |
| | GPT-4(pass@1) | 89.0% | 87.3% | 49.3% | 16.6% | 33.3% | 68.6% | 57.4% |
| | Deepseek-V3(pass@1) | 95.9% | 88.3% | 51.1% | 32.6% | 37.0% | 71.6% | 62.8% |
| | OptiMUS (GPT-4o-2024-05-13) | 78.8% | 77.0% | 43.6% | 20.2% | 31.0% | 45.8% | 49.4% |
| | Qwen2.5-32B(pass@1) | 92.7% | 82.2% | 44.6% | 9.3% | 16.0% | 47.6% | 48.7% |
| **Fine-tuning** | ORLM-Llama-3-8B (reported) | 85.7% | 82.3% | 37.4% | * | 38.0% | * | 60.9% |
| | ORLM-Llama-3-8B (reproduced) | 84.5% | 74.9% | 34.1% | 2.6% | 24.0% | 51.1% | 45.2% |
| | OptMATH-Llama3.1-8B (pass@1) | 55.5% | 73.9% | 40.8% | 24.4% | 18.0% | 55.5% | 44.7% |
| | OptMATH-Qwen2.5-7B (pass@1) | 94.7% | 86.5% | 51.2% | 24.4% | 20.0% | 57.9% | 55.8% |
| | **OptMATH-Qwen2.5-32B (pass@1)** | **95.9%** | **89.9%** | **54.1%** | **34.7%** | **31.0%** | **66.1%** | **62.0%** |
| **Pass@8** | OptMATH-Llama3.1-8B | 97.6% | 94.2% | 71.6% | 51.6% | 37.0% | 66.6% | 69.8% |
| | OptMATH-Qwen2.5-7B | 98.4% | 94.5% | 72.5% | 56.0% | 38.0% | 68.1% | 71.3% |
| | OptMATH-Qwen2.5-32B | 97.9% | 93.9% | 75.4% | 67.4% | 47.0% | 76.8% | 76.4% |

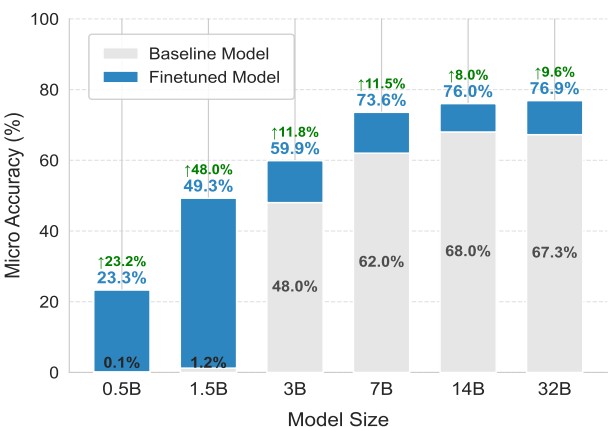

*Figure 6.* Scaling behavior of Qwen2.5 models (0.5B-32B).

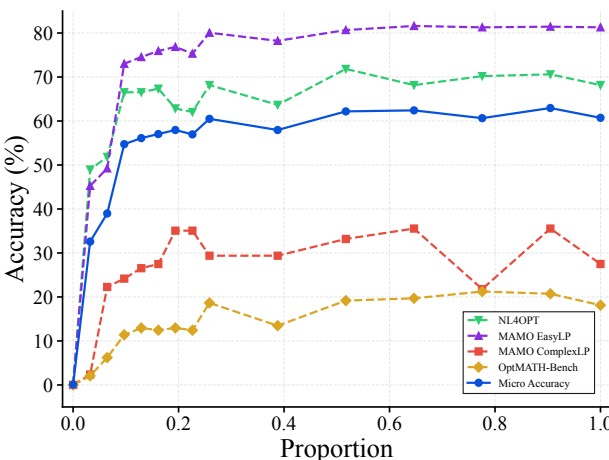

*Figure 7.* Accuracy of Qwen2.5-1.5B within one training epoch.

Detailed descriptions of these benchmarks can be found in Appendix A.1. We use pass@1 accuracy as the evaluation metric, which specifically measures whether the optimal value obtained by the generated code matches the ground truth provided in the benchmark. The detailed matching criteria are described in Appendix A.1. Notably, since prompt design can significantly impact model performance, we maintain consistency by using the same prompt template across all model evaluations (see Appendix E.2 for details). Additionally, comprehensive details about our fine-tuning procedure are provided in Appendix D.3.

**Main Results.** The primary results are presented in Table 1. First, our best-performing model, OptMATH-Qwen2.5-32B, achieves superior performance across all benchmarks, surpassing proprietary large language models such as GPT-3.5-Turbo(Brown et al., 2020a) and GPT4(OpenAI et al., 2023), and reaching a level comparable to Deepseek-V3(Liu et al., 2024a), despite these models often having significantly more parameters. Furthermore, our OptMATH-Qwen2.5-7B out-

performs ORLM-Llama-3-8B, a model of comparable size, demonstrates performance only marginally inferior to GPT-4. Collectively, these results demonstrate that training with OptMATH-Train significantly enhances the model's optimization modeling capabilities.

The results in Table 1 also highlight that different base models possess varying inherent modeling capabilities. For instance, the Llama3.1-8B(Dubey et al., 2024) base model exhibits very poor initial modeling ability, with performance close to 0% on most datasets. However, after fine-tuning with our dataset (OptMATH-Llama3.1-8B), its modeling capability is substantially improved. Despite this significant enhancement, the fine-tuned OptMATH-Llama3.1-8B still performs below the level of OptMATH-Qwen2.5-7B, underscoring the impact of the base model's architecture or pre-training on final fine-tuned performance. Moreover, the strong pass@8 results achieved by the fine-tuned models suggest a high upper limit to their modeling capabilities,

indicating potential for further enhancement through reinforcement learning methods, similar to approaches like Deepseek-R1(Guo et al., 2025).

**Ablation Study on Model Size.** To investigate the effectiveness of OptMATH training across different model scales, we conducted experiments using Qwen2.5 models ranging from 0.5B to 32B parameters. Due to computational constraints, we used a randomly sampled subset of 100,000 training examples. As shown in Figure 6, all models exhibit substantial performance improvements after fine-tuning with OptMATH-Train. Notably, we observe that while larger models generally achieve better absolute performance, the relative performance gains from OptMATH-Train training demonstrate diminishing returns as model size increases.

**Ablation Study on Data Size.** Figure 7 shows how the amount of training data affects Qwen2.5-1.5B's performance across different benchmarks. We observed significant improvements in the model's optimization modeling capabilities even with only a small fraction of the OptMATH-Train dataset. As we gradually increased the size of the training data, the performance gains became less pronounced, exhibiting a typical pattern of diminishing returns. Larger models exhibit smoother learning curves, while smaller models demonstrate greater sensitivity to additional training data, indicating higher potential for improvement through data scaling (detailed results across model sizes can be found in the Appendix D.4).

## 7. Conclusion

In this paper, we introduce a bidirectional data synthesis framework for optimization modeling. It utilizes a two-step process: reverse data generation, where LLMs refine themselves in a loop to create diverse datasets, and auto-formulation, where a specialized model translates natural language into mathematical representations. Our evaluation on NL4OPT, MAMO and OptMATH-Benchmarks demonstrated AutoFormulator's superior performance in generating accurate and well-formed optimization models compared to baseline approaches.

## Acknowledgments

The computational resources were supported by the Center for Intelligent Computing and Song-Shan Lake HPC Center (SSL-HPC) in Great Bay University, Dongguan, China. This work was supported in part by National Key Research and Development Program of China under the grant numbers 2024YFA1012901 and 2024YFA1012903, and the National Natural Science Foundation of China under the grant numbers 12331010 and 12288101. We also thank the anonymous reviewers for their valuable feedback.

## Impact Statement

This study introduces OptMATH, a dataset for optimization modeling, comprising a large-scale training set (OptMATH-Train) and a challenging benchmark (OptMATH-Bench). OptMATH has the potential to democratize optimization by enabling those without expertise to translate real-world problems into mathematical formulations. The OptMATH-Train dataset will significantly improve LLMs' ability to understand and model optimization problems. Furthermore, OptMATH's structured data facilitates the integration of optimization with advanced AI techniques like reinforcement learning, Monte Carlo Tree Search. Additionally, OptMATH-Bench provides a standardized benchmark for evaluating optimization modeling systems, pushing the boundaries of LLM capabilities. Ultimately, OptMATH can improve efficiency and decision-making across industries.

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

# A. Dataset

## A.1. An Introduction of Different Benchmarks

We evaluated the modeling capabilities of our trained model on NL4OPT, MAMO, IndustryOR, OptiBench, and our self-constructed dataset, OptMATH-Bench. Both MAMO and OptMATH-Bench have ground truth annotations, while the original NL4OPT dataset lacks ground truth. To address this, we utilized a LLM to generate initial ground truth for NL4OPT, followed by expert validation and correction for each data. As a result, we obtained the ground truth for the NL4OPT dataset. In addition, we have also analyzed these datasets in terms of problem scenarios and problem model types, and the distribution of scenarios for each dataset is shown in Figure 8, the distribution of problem types for each dataset is shown in Figure 4.

| Scenario | NL4OPT | MAMO EasyLP | MAMO ComplexLP | OptMATH-Bench |
|---|---|---|---|---|
| Agriculture | 6.12% | 4.60% | 2.37% | 0.00% |
| Aviation | 0.00% | 0.00% | 0.95% | 7.25% |
| Construction | 0.82% | 6.60% | 0.47% | 0.52% |
| Education | 0.82% | 4.91% | 0.00% | 0.00% |
| Energy | 1.22% | 4.60% | 2.37% | 3.63% |
| Environment | 0.00% | 5.83% | 0.00% | 0.00% |
| Finance | 2.04% | 10.74% | 6.64% | 5.70% |
| Healthcare | 12.65% | 5.06% | 3.79% | 0.00% |
| Logistics | 6.53% | 1.53% | 14.69% | 12.95% |
| Manufacturing | 31.84% | 4.91% | 5.21% | 36.27% |
| Marketing | 1.22% | 6.44% | 0.00% | 0.00% |
| Military | 0.00% | 5.67% | 0.00% | 0.52% |
| Retail | 4.08% | 5.06% | 3.32% | 0.00% |
| Services | 10.20% | 10.89% | 2.37% | 4.15% |
| Sports | 0.00% | 4.75% | 0.00% | 0.00% |
| Supply Chain | 0.82% | 5.06% | 27.01% | 7.77% |
| Technology | 0.00% | 0.92% | 4.27% | 0.52% |
| Telecommunications | 0.00% | 3.68% | 1.42% | 10.88% |
| Transportation | 16.73% | 7.67% | 19.43% | 5.18% |
| Other | 4.90% | 1.07% | 5.69% | 4.66% |

Figure 8. Scenarios distribution of the datasets.

**NL4OPT**(Ramamonjison et al., 2021) is a curated dataset derived from the NL4OPT Competition, where participants were tasked with developing automated methods to convert natural language problem descriptions into solver-ready code. This dataset primarily focuses on LP (Linear Programming) problems across various contexts, though the underlying mathematical models are relatively uniform, with more complex MIPS (Mixed Integer Programming and Scheduling) problems notably absent. For our experiments, we selected the test set from this dataset, filtered out low-quality examples, and retained a total of 245 high-quality instances.

**MAMO**(Huang et al., 2024) introduces a novel optimization dataset to assess the modeling capabilities of LLMs. The dataset is divided into two main components, *Easy_LP* and *Complex_LP*, containing 652 and 211 instances, respectively. These components cover both LP and MILP problems, capturing a wide range of real-life scenarios. However, the dataset does not include any nonlinear programming (NLP) problems.

**IndustryOR**(Tang et al., 2024) is the first industrial benchmark for operations research, consisting of 100 real-world OR problems collected from eight different industries. This dataset covers five types of optimization problems: linear programming, integer programming, mixed integer programming, non-linear programming, and others, with problems distributed across three levels of difficulty.

**OptiBench**(Yang et al., 2025) introduces a comprehensive benchmark designed to evaluate LLMs' end-to-end solving capabilities in optimization tasks. The dataset consists of 605 carefully selected and manually verified problems, covering linear and nonlinear programming with both integer and mixed integer variables. Since this benchmark does not directly include an "answer" field, we extracted the optimal values to serve as golden answers using both rule-based methods and LLM-based approaches.

**OptMATH-Bench.** As shown in Table 1, while our fine-tuned model achieves remarkable performance on NL4OPT and MAMO_EasyLP, these datasets alone are insufficient to comprehensively evaluate the model's optimization modeling capabilities. Moreover, both NL4OPT and MAMO datasets are limited to linear programming problems, making them less representative of the broader optimization landscape. To address this limitation, we constructed **OptMATH-Bench**, a more challenging dataset for large models that also expands the diversity of problem types.

The creation of OptMATH-Bench followed two distinct routes. The first pathway began with instances initially rejected by our AutoFormulator due to failing the optimal value (OV) check, indicating their inherent difficulty. A "LLM-Committee" (inspired by (Zhao et al., 2024), utilizing diverse powerful models such as GPT-4, Claude, Gemini, and DeepSeek) then filtered these problematic instances. PD and NL pairs were retained only if at least one, but no more than two, committee members successfully formulated them (i.e., passed the OV check). This process isolated problems that were well-posed yet presented non-trivial modeling challenges. Crucially, human OR experts subsequently validated the correctness of these selected pairs and further refined them based on relevance and clarity. The second pathway involved OR experts directly curating challenging problems from external operations research literature, including journals and textbooks. This approach ensured methodological and source independence in the dataset's composition and allowed for the inclusion of known hard problem types, such as NLP and SOCP (as illustrated in Figure 4).

This dataset includes a carefully curated selection of representative mathematical optimization problems that span a broad range of application scenarios, covering LP, MILP, IP, NLP, SOCP, and other common optimization problems. Additionally, the problems in OptMATH-Bench are inherently challenging, making them effective in distinguishing the modeling capabilities of the model.

During evaluation, we observed that certain ambiguities in problem statements could cause the LLM to struggle in determining whether a variable is integer or continuous. To address this, we applied a rule-based substitution approach: as long as the optimal solution derived under either assumption (integer or continuous variable) matches the ground truth, we consider it a pass. We also implemented several other rule-based substitutions to handle variations in solution formats and representations, ensuring a fair evaluation process where all models are assessed using the same standardized evaluation methodology. To determine whether the optimal values are equivalent, we use the following formula:

$$\frac{|y_{pred} - y_{label}|}{|y_{label}| + 1} < \epsilon,$$

where $\epsilon$ is set to 1e-6.

### A.2. Seed Classes

Our seed problem classes were curated by drawing from MIPLIB instances and integrating insights from both Chain-of-Experts(Xiao et al., 2024) and peer-reviewed literature. For each instance, we conducted an in-depth analysis of its structure, starting from the problem description to identify its broader optimization category and further refining it into specific subclasses. To ensure theoretical accuracy, we consulted literature that provided detailed descriptions of these optimization subclasses. Based on these references, we formulated the mathematical representation of each subclass, systematically outlining sets (where applicable), parameters, decision variables, objective functions, and constraints. This step aimed to establish an abstract mathematical framework rather than focusing on specific instances.

We organized comprehensive metadata for each problem class in a structured `metadata.json` file, encompassing subclass names, references, reference links, and LaTeX-formulated mathematical expressions. An example of this metadata structure is provided in Appendix E.5. This systematic documentation not only ensures clarity but also facilitates dataset utilization and future extensions.

Next, we focused on generating new problem instances. We implemented a custom Python class, `Generator()`, in `generator.py`, which contained a step-by-step algorithm to create instances of the identified subclasses (an example is

provided in Appendix E.6). The input parameters and outputs were explicitly defined, with detailed specifications for each parameter's type and valid range documented in `README.md`. We validated the generator by running `test_generator()` with default parameters to ensure the produced instances were both mathematically valid and practically meaningful.

Through this systematic and meticulous approach, we constructed a high-quality dataset of Problem Description (PD) generators that lays a solid foundation for generating natural language descriptions of optimization problems through Backtranslation Pipeline. This dataset is designed to be versatile and scalable, making it suitable for a wide range of applications in optimization research and practice.

| Main Class | Problem Class | Num | Reference |
|---|---|---|---|
| Assignment and Resource Allocation Optimization | Car Selection Problem | 1 | (Burkard et al., 2012) |
| | Contract Allocation Problem | 1 | (Wikipedia, 2025) |
| | Assignment Problem | 2 | (Kuhn, 1955) |
| | Structure-Based Assignment Problem | 1 | (Burkard et al., 2012) |
| | Team Formation Problem | 1 | (Gurobi Optimization, 2025) |
| | Military Personnel Deployment Problem | 1 | (Brown et al., 2004) |
| Combinatorial Optimization | Knapsack Problem | 1 | (Martello & Toth, 1990) |
| | Market Share Optimization Problem | 1 | (Cooper, 1993) |
| | Set Multi-Cover Problem | 1 | (Winston, 2004) |
| | Set Cover Problem | 1 | (Caprara et al., 2000) |
| Cutting and Packing Optimization | Bin Packing Problem | 1 | (Garey & Johnson, 1981) |
| | Blending Problem | 1 | (Zieyel, 1988) |
| | Cutting Stock Problem | 1 | (Gilmore & Gomory, 1961) |
| Domain-Specific Optimization | Diet Problem | 3 | (Garille & Gass, 2001) |
| | Unit Commitment Problem | 1 | (Padhy, 2004) |
| | Farm Planning Problem | 1 | (Hazell & Norton, 1986) |
| Facility Location Optimization | Facility Location Problem | 2 | (Toregas et al., 1971) |
| | Capacitated Facility Location Problem | 2 | (Daskin, 1995) |
| | Transportation Problem, Airline Industry Resource Allocation | 1 | (Toth, 2002) |
| | Facility Dispersion Problem | 2 | (Kuby, 1988) |
| Financial and Revenue Optimization | Portfolio Optimization Problem | 1 | (Markowitz, 1952) |
| | Profit Maximization Problem | 1 | (Hillier & Lieberman, 2014) |
| | Revenue Management Problem | 1 | (Chen et al., 2022) |
| | Revenue Maximization Problem | 1 | (Weatherford & Bodily, 1997) |
| Network Flow Optimization | Multi-Commodity Capacitated Network Design Problem | 1 | (Gendron et al., 1999) |
| | Multi-Commodity Transportation Problem | 1 | (Smith, 1994) |
| | Minimum Cost Flow Problem | 1 | (Klein, 1967) |
| | Multi-Commodity Network Flow Problem | 1 | (Smith, 1994) |
| | Network Flow Problem | 1 | (Ford & Fulkerson, 1956) |
| | Static Line Planning Problem | 1 | (Schöbel, 2012) |
| | Supply Chain Optimization | 1 | (Cordeau et al., 2006) |
| | Network Optimization | 1 | (Bertsekas, 1998) |
| Production Planning and Scheduling Optimization | Capacitated Lot-Sizing Problem | 1 | (Florian & Klein, 1971) |
| | Factory Planning Problem | 1 | (Pinedo, 2022) |
| | Flow Shop Scheduling Problem | 1 | (Rajendran, 1995) |

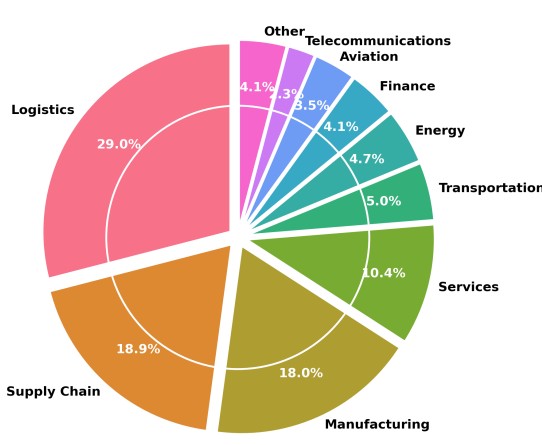

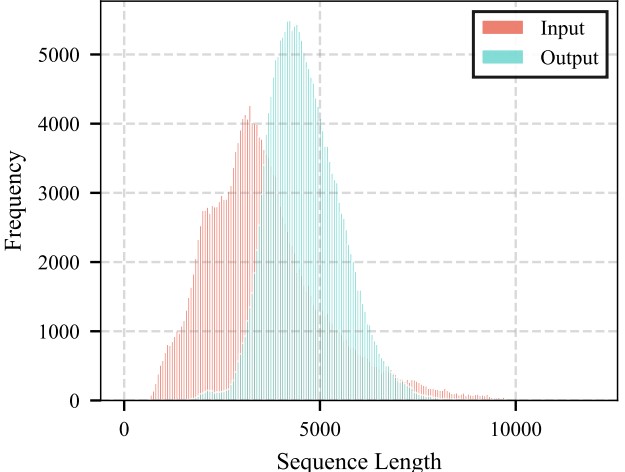

*Figure 9.* Distribution of Application Scenarios across OptMATH-Train

*Figure 10.* Sequence Length Distribution of OptMATH-Train

| Main Class | Problem Class | Num | Reference |
|---|---|---|---|
| | Job Shop Scheduling Problem | 1 | (Adams et al., 1988) |
| | Discrete Lot-Sizing and Scheduling Problem | 1 | (Fleischmann, 1990) |
| | Production Planning Problem | 3 | (Herrmann, 2006) |
| | Lot-Sizing Problem | 1 | (Drexl & Kimms, 1997) |
| Transportation and Routing Optimization | Aircraft Assignment Problem | 1 | (Abara, 1989) |
| | Aircraft Landing Problem | 1 | (Beasley et al., 2000) |
| | Transportation Problem | 2 | (SteadieSeifi et al., 2014) |
| | Traveling Salesman Problem | 1 | (Dantzig et al., 1954) |
| | Operations Optimization | 1 | (Golden et al., 2008) |
| | Capacitated Vehicle Routing Problem with Time Windows | 3 | (Solomon, 1987) |

### A.3. OptMATH-Train

The OptMATH-Train dataset consists of over 150k reverse-generated samples and 50k augmented instances, forming a comprehensive collection of optimization problems. The dataset encompasses a rich variety of real-world application scenarios. As illustrated in Figure 9, the dataset covers over 10 major application domains spanning across both core business sectors and specialized industries, demonstrating extensive coverage of real-world optimization scenarios. The substantial proportions in logistics, supply chain, and manufacturing ensure robust representation of primary industrial applications, while the balanced inclusion of sectors like transportation, energy, and finance provides comprehensive coverage of specialized use cases. This thoughtful allocation of problems across different domains not only prevents data concentration but also maintains sufficient samples for each sector, enabling effective model training and evaluation.

The sequence length distribution of OptMATH-Train, as shown in Figure 10, exhibits a well-balanced profile for both input and output sequences. The distribution approximates a normal distribution with mild right-skewness, centered around 5,000 characters, demonstrating a natural variation in problem complexity. This balanced distribution pattern is particularly advantageous for model training, as it ensures sufficient context length for complex problem representation while maintaining computational efficiency. Furthermore, the moderate right-skewness encompasses challenging cases with extended sequences, which is essential for developing robust models capable of handling sophisticated optimization problems that require comprehensive reasoning and detailed solution steps.

# B. Details of Instance Generation

## B.1. An Example for Measuring the Complexity

Let binary variables $y_1, y_2 \in \{0, 1\}$ indicate whether products 1 and 2 are produced, integer variables $x_1, x_2 \in \mathbb{Z}^+$ represent production quantities, and a continuous variable $z \geq 0$ denote total cost. The objective function minimizes total operational costs: $\min z + 10y_1 + 8y_2$. The constraints span four categories: First, linear constraints include the resource limitation $2x_1 + 3x_2 \leq 100$ and market demand bounds $x_1 \leq 50$, $x_2 \leq 30$. Second, indicator constraints using the Big-M method (with $M = 100$) enforce minimum production levels when activated: $y_1 = 1 \implies x_1 \geq 5$ is reformulated as $x_1 \geq 5 - 100(1 - y_1)$, and analogously for $y_2 = 1 \implies x_2 \geq 3$. Third, a quadratic constraint $z \geq 0.5x_1^2 + 0.3x_2^2$ integrates inventory costs into the objective. Finally, a general nonlinear constraint $x_1 e^{x_2} \leq 100$ captures efficiency coupling between products.

To compute the complexity score $S(\text{PD})$, we identify: 2 binary variables, 2 integer variables, and 1 continuous variable; 3 linear constraints, 2 indicator constraints, 1 quadratic constraint, and 1 nonlinear constraint. The Big-M factor frequency is $f_{\text{BigM}} = 2$, while the average number of terms per expression $\overline{L_{\text{expr}}} \approx 2.71$ is derived from structural analysis of constraints and the objective. With all weights set to 1, the total complexity score becomes $S = 16.71$. This example demonstrates how modeling choices (e.g., introducing nonlinear terms, Big-M parameterization) directly influence the score, providing a quantitative framework for assessing model complexity.

## B.2. An Overview of the Generated LP Files

The average lengths of LP files for different difficulty levels are illustrated in Figure 11. We define the complexity thresholds for the five difficulty levels—easy, medium_easy, medium, medium_hard, and hard—as [25, 75], [50, 100], [75, 125], [100, 150], and [125, 175], respectively. The results demonstrate that our feedback-driven problem data generation approach is effective. The average length of the generated LP files across the five difficulty levels is presented in Figure 12, categorized by seed data name. All generated LP files are feasible and possess optimal solutions.

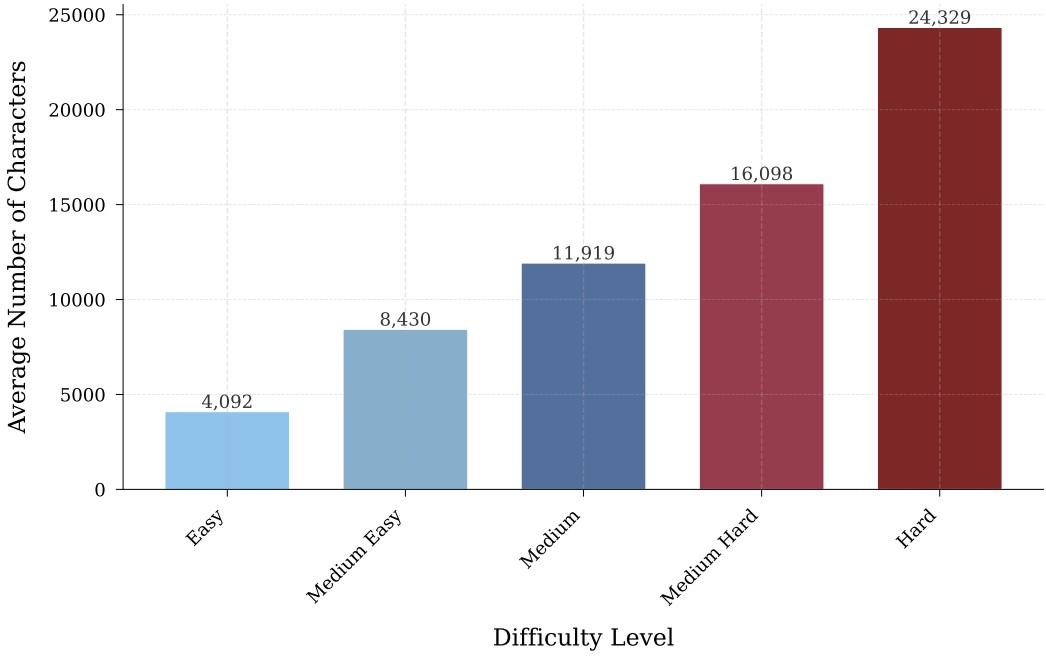

*Figure 11.* Distribution of LP file lengths across generated instances by difficulty levels.

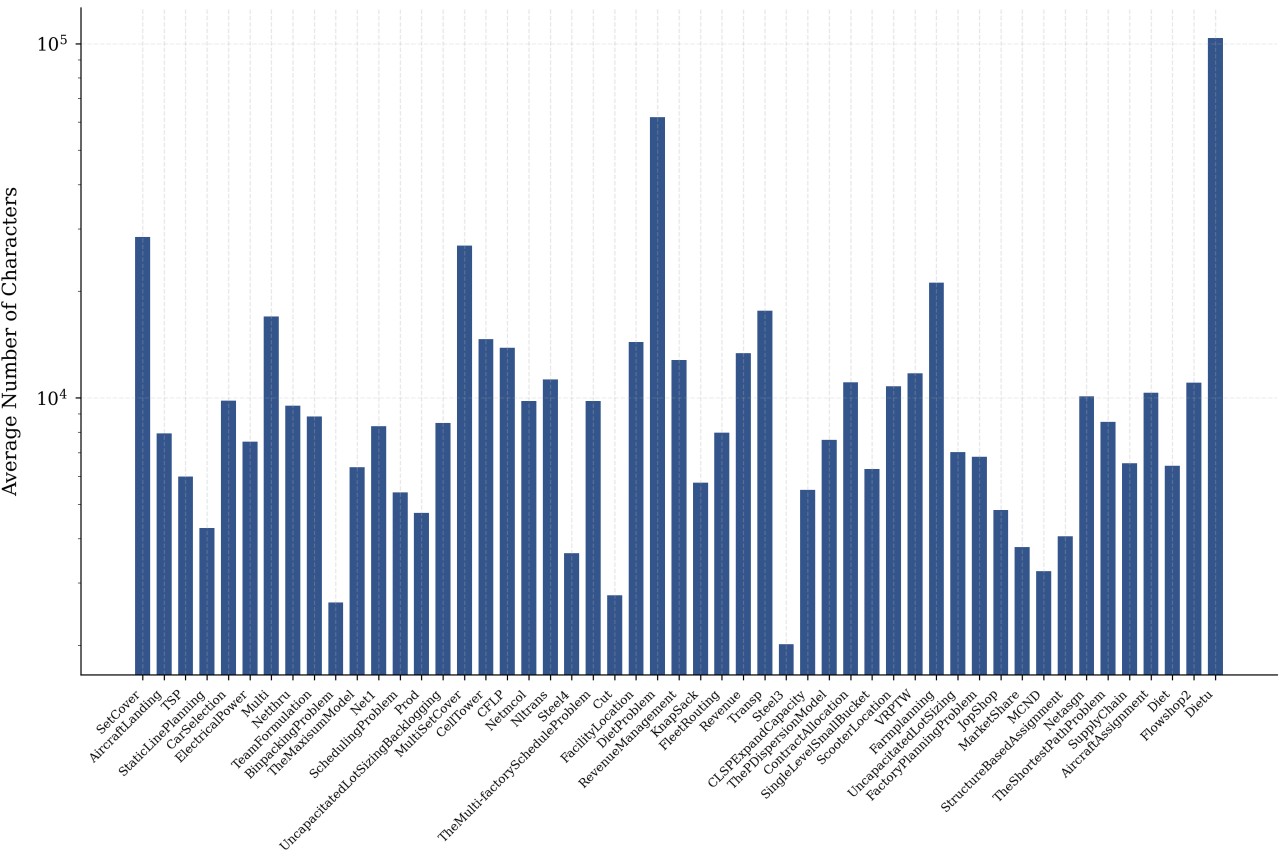

*Figure 12.* Distribution of LP file lengths across generated instances by problem types.

## C. Details of Backtranslation

### C.1. Backtranslation Pipeline

In our reverse generation pipeline, we employ Deepseek-V3(Liu et al., 2024a) as our foundation model and configure its temperature parameter to 0.8 to enhance the diversity of generated problems. Furthermore, to achieve rich contextual diversity, we implement a random scenario assignment mechanism during the Initial Generate phase. This mechanism directs the LLM to synthesize problems that optimally integrate the mathematical characteristics with the designated scenario context. The detailed prompt is elaborated in Section E.1.

Through this backtranslation process, we initially generated approximately 120,000 easy optimization problems. As shown in Figure 15, the length distribution of problem descriptions exhibits a right-skewed pattern, with most problems containing 2,000 to 5,000 characters. After applying rejection sampling, around 40% of the generated problems were filtered out while maintaining a similar distribution pattern. This consistency in distribution before and after filtering suggests that our quality control process effectively removes low-quality samples without introducing length-related biases, ensuring the retained problems maintain natural and appropriate descriptive lengths. After multi-stage refinement including semantic verification and difficulty calibration, the pipeline ultimately produced 150,000 rigorously validated optimization problems. Together with 50,000 augmented instances, this curated collection forms our OptMATH-Train dataset, where each instance demonstrates: (1) Contextual alignment between mathematical formulations and real-world scenarios, (2) Controllable complexity levels matching specified difficulty tiers, and (3) Natural language expressions adhering to authentic problem-solving discourse patterns. The hierarchical quality assurance framework ensures the dataset's applicability for both educational interventions and benchmarking mathematical reasoning systems.

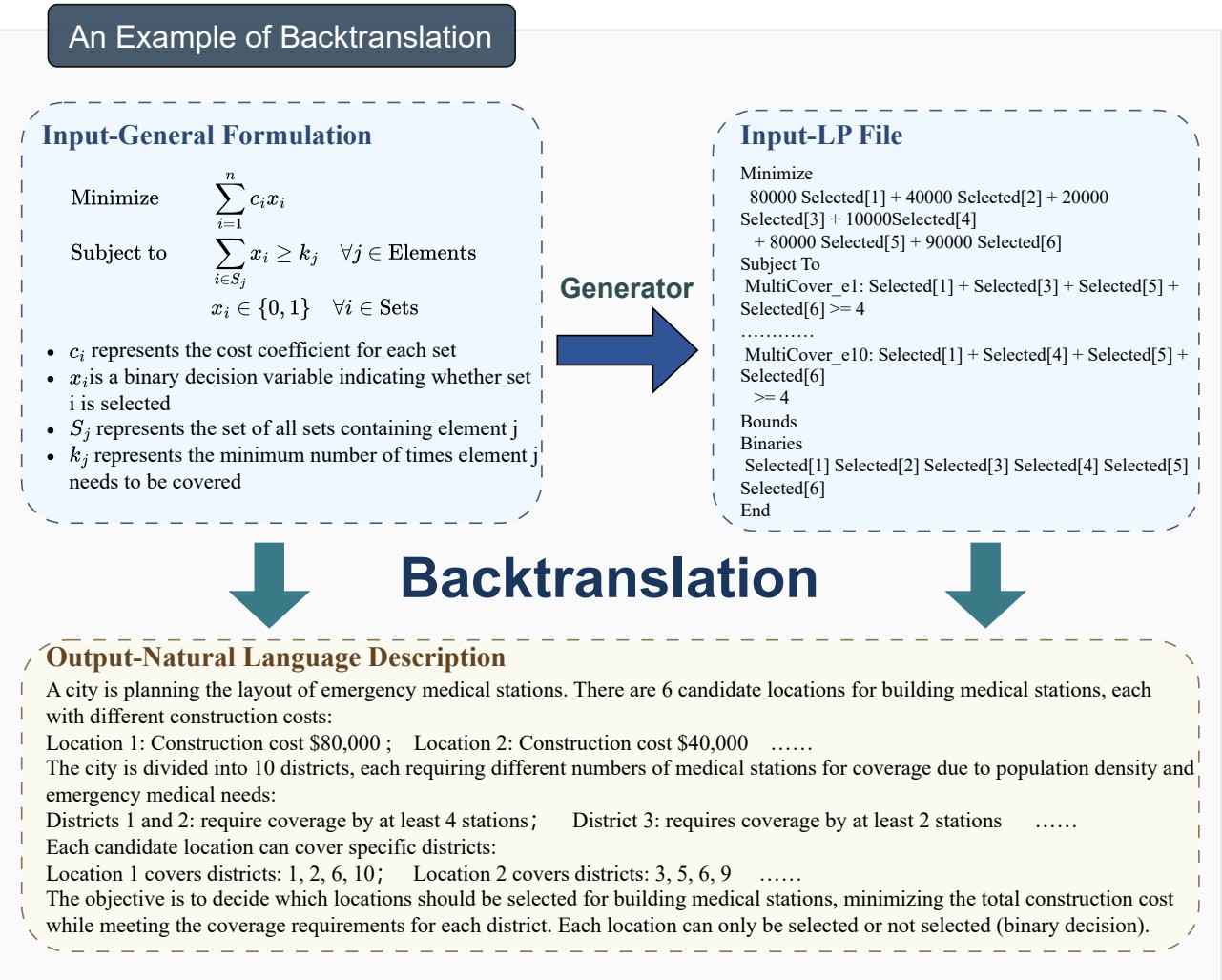

*Figure 13.* An example of backtranslation: transforming mathematical formulations and LP files into natural language descriptions of optimization problems. The process transforms formal mathematical notation and concrete data into human-readable problem descriptions.

## C.2. Ablation Study on the Impact of Self-Refine Iterations

To validate the effectiveness of each step in our backtranslation pipeline, we conducted comprehensive ablation studies. We first compared the accuracy between using only the Generate step versus implementing the complete pipeline with Generate, Self-criticize, and Self-refine steps. To investigate the impact of parameter $T$ on the acceptance rate of rejection sampling, we randomly selected 500 instances for evaluation, with results shown in Figure 14. The results demonstrate that our Self-Refine loop ($T \geq 1$) consistently outperforms direct generation ($T = 0$) in terms of acceptance rate. While there are some fluctuations in performance across different $T$ values, possibly due to the inherent hallucination tendencies of large language models, we observe that setting $T = 1$ achieves a satisfactory acceptance rate of 61.56%. Considering the trade-off between performance and computational efficiency (token usage), we adopt $T = 1$ in our final data synthesis process.

## D. Details of AutoFormulation

### D.1. CoT Instructions of AutoFormulation

To support comprehensive mathematical modeling capabilities, we developed a diverse set of CoT instructions, which are detailed in Section E.3. These instructions vary in their decomposition approaches, intermediate reasoning steps, and

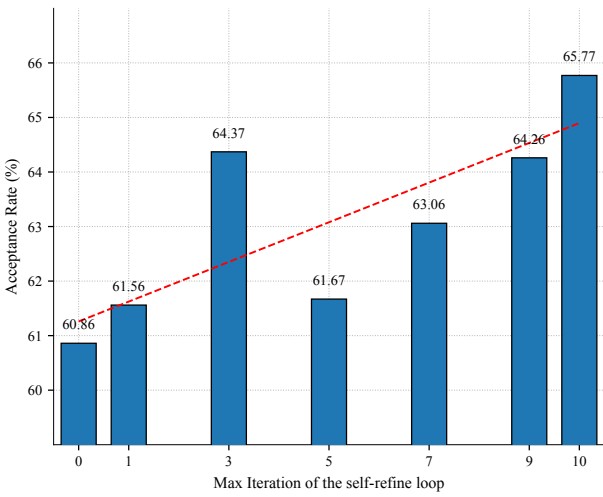

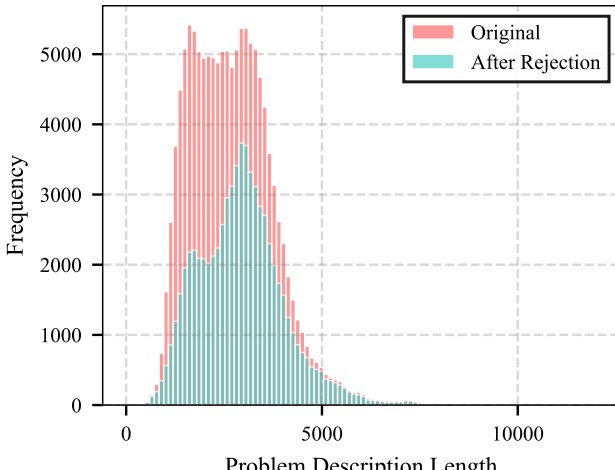

*Figure 14.* Acceptance Rate vs. Maximum Iteration of Self-Refine Loop. The bar chart illustrates the acceptance rate achieved at different maximum iteration limits.

*Figure 15.* Distribution of Natural Language Description Lengths for Easy Problems in OptMATH-Train Dataset. The histogram compares the length distribution before and after rejection sampling, showing the quality filtering process.

presentation formats, providing multiple pathways for problem formulation. In Figure 16, we present one representative formulation pattern from our instruction set. This format includes three key components: a general mathematical formulation with standard notation, a detailed instance-specific formulation with complete parameter specifications, and the corresponding Python implementation using Gurobi. However, this represents just one of many possible formulation styles. Other formats in our instruction set may use different ordering of steps, alternative notation systems, or various levels of mathematical abstraction. The diversity in formulation patterns ensures that our dataset captures a wide range of valid mathematical modeling approaches while maintaining logical coherence and mathematical correctness.

### D.2. Ablation Study on Augmentation

As mentioned in the previous section, the purpose of data augmentation is to increase the diversity of the dataset and generate more non-standard problems, which can help the fine-tuned model to solve more difficult problems. We use 50k raw data, augmented data and mixed data (50% raw data and 50% augmented data) for fine-tuning training on the Qwen2.5-7B model, respectively, and the results show in Table 3 that the model fine-tuned with raw data performs better in solving relatively simple problems, augmented data performs better in solving difficult problems, and mixed data combines the advantages of the above two very well, being the best in terms of average accuracy across the four types of test sets.

*Table 3.* Comparison of Original Data and Augmentation Data in Training Models.

| Types | NL4OPT | MAMO EasyLP | MAMO ComplexLP | OptMATH Bench | Micro Avg | Macro Avg |
|---|---|---|---|---|---|---|
| Without Augmentation | 86.9% | **88.0%** | 44.5% | 31.1% | 72.1% | 62.3% |
| Without Original | 82.9% | 85.5% | 44.7% | 23.6% | 69.2% | 59.2% |
| Mixture of Augmentation and Original | **87.3%** | 87.7% | **48.1%** | **33.3%** | **73.1%** | **64.1%** |

### D.3. SFT

We employ the LlamaFactory framework for fine-tuning (Zheng et al., 2024). We select the Qwen2.5 series (0.5B∼32B) as our base models (Yang et al., 2024), and the hyperparameters are generally set as follows: initial learning rate of 1e-4, 1∼3 epochs, LoRA rank of 32, LoRA alpha of 32, and LoRA dropout of 0.1. While there are minor variations in hyperparameters

## An Example of AutoFormulation

### Input-Natural Language Description

A city is planning the layout of emergency medical stations. There are 6 candidate locations for building medical stations, each with different construction costs:

Location 1: Construction cost \$80,000 ;   Location 2: Construction cost \$40,000   ……

The city is divided into 10 districts, each requiring different numbers of medical stations for coverage due to population density and emergency medical needs:

Districts 1 and 2: require coverage by at least 4 stations；    District 3: requires coverage by at least 2 stations    ……

Each candidate location can cover specific districts:

Location 1 covers districts: 1, 2, 6, 10；   Location 2 covers districts: 3, 5, 6, 9   ……

The objective is to decide which locations should be selected for building medical stations, minimizing the total construction cost while meeting the coverage requirements for each district. Each location can only be selected or not selected (binary decision).

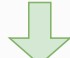 **AutoFormulation** 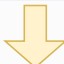

### Output-General Formulation

$$\text{Minimize} \quad \sum_{i=1}^{n} c_i x_i$$

$$\text{Subject to} \quad \sum_{i \in S_j} x_i \geq k_j \quad \forall j \in \text{Elements}$$

$$x_i \in \{0, 1\} \quad \forall i \in \text{Sets}$$

- $c_i$ represents the cost coefficient for each set
- $x_i$ is a binary decision variable indicating whether set i is selected
- $S_j$ represents the set of all sets containing element j
- $k_j$ represents the minimum number of times element j needs to be covered

### Output-Instance Formulation

**1. Sets**

L = {1,...,6}: set of potential locations

D = {1,...,10}: set of districts

**2. Parameters**

c[i]: construction cost for location i ∈ L

    c[1] = 8, c[2] = 4, c[3] = 2, c[4] = 1, c[5] = 8, c[6] = 9

r[j]: coverage requirement for district j ∈ D

    r[1] = 4, r[2] = 4, r[3] = 2, r[4] = 4, r[5] = 5

    r[6] = 1, r[7] = 2, r[8] = 3, r[9] = 4, r[10] = 4

a[i,j]: coverage matrix, 1 if location i can cover district j, 0 otherwise.

**3. Variables**

- x[i] ∈ {0,1}: binary decision variable
- x[i] = 1 if location i is selected
- x[i] = 0 otherwise

**4. Objective Function**

    Minimize: ∑(i∈L) c[i]×x[i] (total construction cost)

**5. Constraints**

- Coverage requirements: ∑(i∈L) a[i,j]×x[i] ≥ r[j], ∀j∈D
- Binary constraints: x[i] ∈ {0,1}, ∀i∈L

### Output-Python Code with Gurobi

```python
from gurobipy import *

# Create a new model
m = Model("Medical Station Location")

# Parameters
costs = {
1: 8, # $80,000
2: 4,
…………
}

# Coverage requirements for each district
coverage_req = {
1: 4,
2: 4,
3: 2,
…………
}
# Coverage matrix
coverage = {
1: [1, 2, 6, 10],
2: [3, 5, 6, 9],
…………
}
# Decision Variables
selected = m.addVars(costs.keys(),
        vtype=GRB.BINARY, name="Selected")

# Objective Function
m.setObjective(quicksum(costs[i] * selected[i] for
        i in costs.keys()), GRB.MINIMIZE)

# Constraints
for district in coverage_req.keys():
        # Sum up all locations
        m.addConstr(quicksum(selected[loc]
        for loc in coverage.keys() if district in
        coverage[loc])>=coverage_req[district],
        f"MultiCover_e{district}")

# Optimize model
m.optimize()

# Print results
if m.status == GRB.OPTIMAL:
    print("\nOptimal solution found:")
else:
    print("No solution found")
```

*Figure 16.* An Example of AutoFormulation

across different experiments, the overall settings remain similar and we omit these details for brevity. Notably, as illustrated in our framework diagram 1, the entire AutoFormulator training process is an iterative cycle. The Rejection Sampling in Step 2 relies on AutoFormulator's modeling capabilities - stronger modeling abilities lead to higher pass rates and better data quality. Similarly, the data augmentation phase depends on AutoFormulator's modeling competence. Higher quality data, in turn, results in a more capable Formulator through training.Through this systematic model fine-tuning and data augmentation approach, we have developed a dynamically evolving fine-tuning framework. This framework not only accurately transforms natural language descriptions into mathematical formulations and solver code but, more importantly, establishes a self-improving data flywheel mechanism. This positive feedback loop enables the AutoFormulator system to continuously enhance its capability in handling complex optimization problems through ongoing learning and self-optimization, forming a virtuous growth cycle.

## D.4. Detailed Ablation Studies on Model Size and Data Size

To investigate the impact of model capacity and training data volume on optimization modeling performance, we conducted two sets of experiments using OptMATH-Train. For the model size study in Figure 6 and Figure 17, we compare the performance of baseline and finetuned Qwen2.5 models ranging from 0.5B to 32B parameters. For the data scaling analysis in Figure 7 and Figure 18, we track the accuracy progression within the first training epoch across different model sizes, using varying proportions of the training data.

The model size experiments reveal distinct scaling patterns across benchmarks. On NL4OPT, the performance improves from 12.7% (0.5B) to 96.7% (32B), showing particularly rapid gains in the 0.5B-3B range. For MAMO EasyLP, we observe similar but more moderate improvements, with accuracy increasing from 31.9% to 90.5%. However, on more challenging benchmarks like MAMO ComplexLP and OptMATH-Bench, even the largest models achieve relatively modest gains, reaching 52.6% and 30.6% respectively at 32B parameters.

The comparison between baseline and OptMATH-Train finetuned models reveals interesting patterns across different model scales. For simpler benchmarks like NL4OPT and MAMO EasyLP, while the performance gap narrows with increased model size, OptMATH-Train finetuning still provides consistent improvements even for the largest models. More notably, on complex benchmarks such as MAMO ComplexLP and OptMATH-Bench, models finetuned on OptMATH-Train demonstrate substantial performance gains across all model sizes, highlighting the effectiveness of our training dataset in enhancing models' capabilities for challenging optimization problems.

The data scaling analysis reveals distinct learning dynamics across model sizes. Smaller models (0.5B, 1.5B, 3B) exhibit higher initial performance variance during training, while larger models (7B, 14B) demonstrate more stable learning curves from the outset. Notably, all model sizes achieve relative performance stability after utilizing approximately 40% of the training data, though the absolute performance levels differ significantly. The 3B model, for instance, maintains consistently higher performance across all benchmarks while requiring a similar proportion of training data to reach stability.

This efficient data utilization pattern holds true across all benchmarks, regardless of their complexity levels. Whether for the relatively straightforward tasks in NL4OPT or the more challenging problems in OptMATH-Bench, models typically converge to their peak performance using around 40% of the available training data. The remaining 60% of the data contributes primarily to fine-tuning and minor performance adjustments rather than substantial improvements.

*Table 4.* Performance comparison of Qwen2.5 models of varying sizes on mathematical optimization tasks. The percentages in parentheses indicate improvements after fine-tuning.

| Models | NL4OPT | MAMO EasyLP | MAMO ComplexLP | OptMATH-Bench | Micro AVG |
|---|---|---|---|---|---|
| Qwen2.5-0.5B | 0.00% | 0.15% | 0.00% | 0.00% | 0.08% |
| Qwen2.5-0.5B(Finetuned) | 12.65% (↑12.65%) | 31.90% (↑31.75%) | 16.59% (↑16.59%) | 15.03% (↑15.03%) | 23.29% (↑23.21%) |
| Qwen2.5-1.5B | 0.00% | 2.15% | 0.95% | 0.00% | 1.23% |
| Qwen2.5-1.5B(Finetuned) | 46.12% (↑46.12%) | 68.10% (↑65.95%) | 22.75% (↑21.80%) | 18.65% (↑18.65%) | 49.27% (↑48.04%) |
| Qwen2.5-3B | 67.35% | 65.18% | 16.11% | 0.52% | 48.04% |
| Qwen2.5-3B(Finetuned) | 68.57% (↑1.22%) | 80.98% (↑15.80%) | 25.59% (↑9.48%) | 15.03% (↑14.51%) | 59.88% (↑11.84%) |
| Qwen2.5-7B | 86.94% | 83.59% | 21.80% | 1.55% | 62.03% |
| Qwen2.5-7B(Finetuned) | 86.94% | 89.42% (↑5.83%) | 48.82% (↑27.02%) | 30.05% (↑28.50%) | 73.56% (↑11.53%) |
| Qwen2.5-14B | 93.47% | 82.52% | 42.65% | 14.51% | 68.02% |
| Qwen2.5-14B(Finetuned) | 95.51% (↑2.04%) | 90.49% (↑7.97%) | 51.18% (↑8.53%) | 29.53% (↑15.02%) | 76.02% (↑8.00%) |
| Qwen2.5-32B | 92.65% | 82.21% | 44.55% | 9.33% | 67.26% |
| Qwen2.5-32B(Finetuned) | 96.73% (↑4.08%) | 88.04% (↑5.83%) | 56.4% (↑11.85%) | 36.27% (↑26.94%) | 76.86% (↑9.60%) |

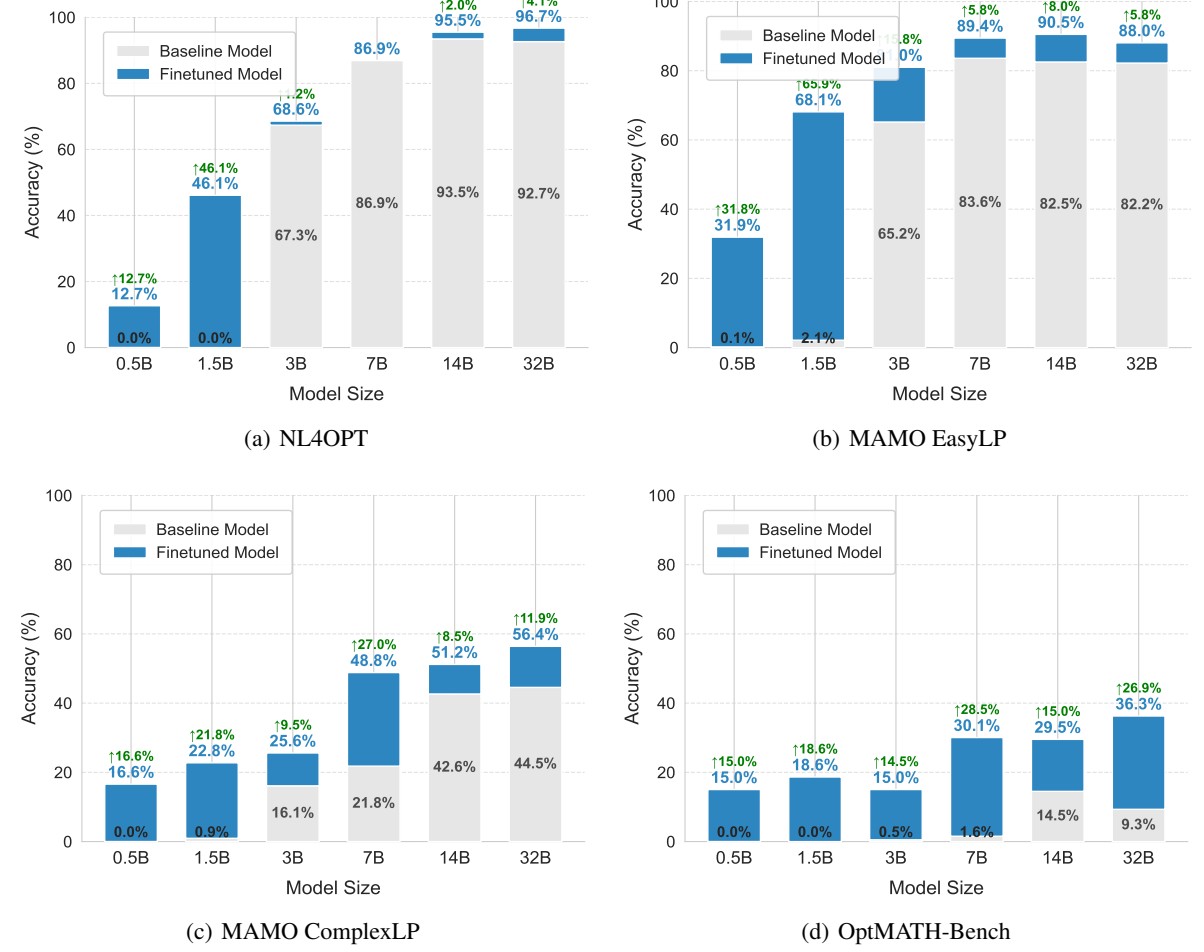

(a) NL4OPT

(b) MAMO EasyLP

(c) MAMO ComplexLP

(d) OptMATH-Bench

*Figure 17.* Scaling behavior of Qwen2.5 models (0.5B-32B) on various benchmarks.

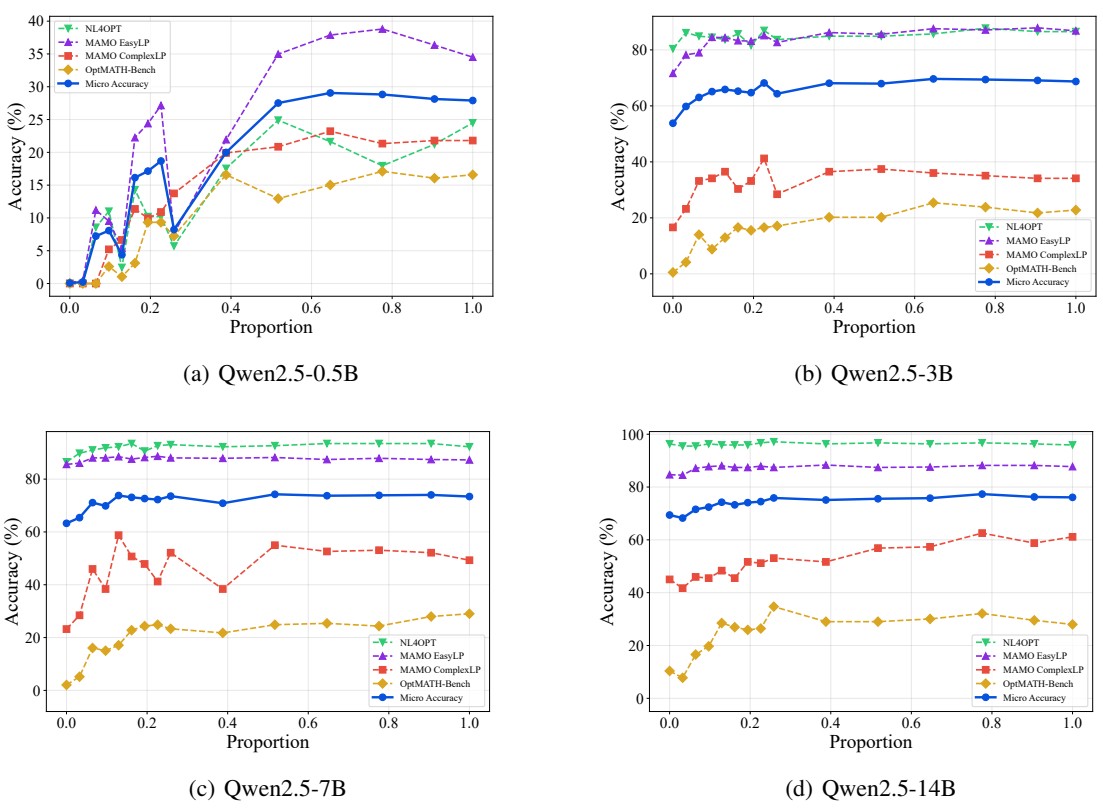

(a) Qwen2.5-0.5B

(b) Qwen2.5-3B

(c) Qwen2.5-7B

(d) Qwen2.5-14B

*Figure 18.* Scaling behavior of Qwen2.5-0.5B, Qwen2.5-3B, Qwen2.5-7B and Qwen2.5-14B Accuracy Within One Training Epoch.

# E. Prompt Templates

In this section, we present all the important prompt templates. Due to space constraints, certain parts are omitted, and only the prompt frameworks are shown.

## E.1. Reverse Data Generate Prompt

GENERATE PROMPT

```
As an Operations Research Expert, analyze the given mathematical optimization
    expression and LP data.

......

Input Mathematical Expression:
{{mathematical_expression}}

Input LP Data:
{{lp_data}}

Reference Examples:
{{examples}}

## Required Output:
Provide ONLY a clear, detailed natural language description of the optimization
    problem that:
- Describes the complete scenario
- States all decisions to be made
- Specifies the objective clearly
- Incorporates all constraints and conditions naturally
- Includes all numerical parameters within narrative
- Uses appropriate domain terminology
- Maintains mathematical accuracy without showing formulation
```

SELF-CRITICISM PROMPT

```
As an Operations Research Expert, evaluate if the generated problem description
    matches the mathematical optimization problem...

Input LP Data:
{{lp_data}}

Generated Problem Description:
{{problem_description}}

Analysis Steps...

## Required Output:
If perfect match:
"Complete Instance"

If inconsistencies exist:
"Incomplete Instance:
[List specific discrepancies...]"
```

SELF-REFINEMENT PROMPT

As an Operations Research Expert, analyze the criticism and refine the problem
    description if needed.

First, check the criticism result:
{{criticism}}

If the criticism shows "Complete Instance":
Output "Nothing need to refine"

Otherwise, follow these steps to generate an improved description:

1. Review Input Materials:
Mathematical Expression:
{{mathematical_expression}}

LP Data:
{{lp_data}}

Initial Description:
{{initial_description}}

2. Task:
Based on the criticism feedback, LP data information, and initial description,
    generate a complete and accurate problem description.

Required Output:
If criticism is "Complete Instance":
Output "Nothing need to refine"

Otherwise:
[Direct natural language description of the optimization problem]
- No introductory phrases or meta-commentary
- No section headers or separators
- Just the complete problem description in clear natural language
- Ensure exact match with all LP data parameters
- Include all constraints and objectives naturally
- Avoid mathematical notation

Note: The output should be ONLY the complete natural language description itself,
    with no additional text or formatting.

## E.2. Baseline Prompt

```
BASELINE PROMPT TEMPLATE FOR OPTIMIZATION MODELING

Below is an operations research question. Build a mathematical model and
    corresponding python code using `gurobipy` that appropriately addresses the
    question.
# Question:
{}

# Notes:
- Please output Python code starting with the following lines:```python\n\nimport
    gurobipy as gp\nfrom gurobipy import GRB\n```
- Make sure the model variable is named `model`.
- Avoid using "<" and ">" in Gurobi constraints; instead, use "<=" or ">=" as
    appropriate.
- Carefully determine whether the variable is an integer or a continuous variable.

# Response:

(Provide your response here,keep the notes above in mind)
```

## E.3. AutoFormulation Instructions

```
CoT INSTRUCTIONS

instructions = [
    # Total instructions: 15 entries
    # Showing 5 representative examples below...

    "Below is an operations research question. Build a mathematical model and
    corresponding Python code using `gurobipy` to solve it.",

    "Create a complete solution that includes: 1) Mathematical formulation 2) Python
     code using gurobipy 3) Results interpretation. Ensure all variables and
    constraints are properly defined.",

    "Transform this operations research problem into a mathematical model and
    implement it in Python with gurobipy. Include clear variable definitions and
    constraint explanations.",

    "The following is an operations research problem. Let's solve it step by step:
    1) Identify the decision variables, objective function, and constraints 2)
    Formulate the mathematical model 3) Implement the solution using Gurobi in
    Python 4) Verify and interpret the results.",

    "This is an operations research problem. Follow this structured approach: Begin
    with understanding the problem -> Identify the key variables -> Analyze the
    constraints -> Develop the mathematical model -> Solve it programmatically using
     Gurobi in Python. Provide clear reasoning and explanations at each stage."
]
```

## E.4. Prompts for Configuration Selecting

---

**INITIALIZING THE PARAMETERS**

```
You are an optimization expert helping to tune the parameters in:
\begin{verbatim}
{generator_code}
\end{verbatim}

CRITICAL REQUIREMENT:
Generated instances MUST be solvable to OPTIMALITY by Gurobi...

Model Complexity Score Calculation:
1. Variable Score:
   - Binary variables: weight = {weights['alpha\_bin']}
   - Integer variables: weight = {weights['alpha\_int']}
   - Continuous variables: weight = {weights['alpha\_cont']}
2. Constraint Score...
3. Additional Complexity Factors...

Total Score = Variable Score + Constraint Score + Big-M Score + Expression Score

Secondary Requirements:
1. Model Complexity: {complexity\_score\_min} to {complexity\_score\_max}
2. Solve Time: {min\_solve\_time} to {max\_solve\_time} seconds

Return parameter values in JSON format matching 'default\_parameters' structure:
- Keep exact same keys
- Preserve data types
- Use lists for tuples
- Format with proper indentation
```

---

**FEEDBACK PROMPT**

```
Based on testing {total_instances} instances with your suggested parameters:
{last_suggested_parameters}

Here are the detailed results:

1. Solution Status Analysis:
   - Total Instances: {total_instances}
   - Solvable Instances: {solvable_instances}
   - OPTIMAL Solutions: {optimal_instances} ({optimal_rate:.1f}%)
   - Solution Status Distribution:
     ...
   - Status Percentages:
     ...

2. Overall Performance (Only OPTIMAL Solutions):
   - Success Rate: {success_rate:.1f}% ({results["requirements_met"]["
    all_requirements"]} out of {total_instances} instances met all requirements)
   - Note: Only OPTIMAL solutions are considered successful

3. Requirements Satisfaction (Only OPTIMAL Solutions):
   Complexity Score Distribution:{analyze_distribution("complexity")}
   - Required range: {requirements}
   - Success rate: {num_satisfying_requirements/total_instances}

   Solve Time Distribution:{analyze_distribution("solve time")}
   - Required range: {requirements}
   - Success rate: {num_satisfying_requirements/total_instances}
```

---

```
4. Model Structure Analysis (Only OPTIMAL Solutions):
   Variable Distributions:
   Binary Variables: ...

   Constraint Distributions:
   Linear Constraints: ...
   Indicator Constraints: ...
   Quadratic Constraints: ...
   General Constraints: ...

   Complexity Score Components: ...

5. Distribution Analysis Insights:
    {throughout analysis of the instances generated by the parameters by calling
    statistics package in Python}

Based on these results and distribution analysis, please suggest parameter values
    that would:
1. MAXIMIZE the proportion of instances that reach OPTIMAL status
2. Adjust the model complexity to meet the target score range (for OPTIMAL instances
    )
3. Maintain solve times within the required range (for OPTIMAL instances)
4. Reduce variability in key metrics where high variance was detected
5. Increase the overall success rate

Return your response in JSON format, strictly following the structure of the
    previous suggestions dictionary. Ensure that:
1. All keys remain exactly the same as in the previous suggestions
2. The data types for each value are preserved
3. The JSON should be properly formatted with indentation for readability
4. Do not add any new keys or remove any existing keys
```

# E.5. An Example of Metadata

---

**Metadata**

**Subclass:** Bin Packing

**Reference:** Garey, M. R. and Johnson, D. S. "Approximation Algorithms for Bin Packing Problems: A Survey." *Analysis and Design of Algorithms in Combinatorial Optimization (1981)*

**Reference URL:** https://doi.org/10.1007/978-3-7091-2748-3_8

**Mathematical Formula:**

*Consider $n$ items, where each item $i$ has:*

- Weight $s_i$: The weight of item $i$

*The problem includes:*

- Bin Capacity $c$: The uniform capacity of each bin

- Bin Usage Variable $y_j$: A binary variable indicating whether bin $j$ is used

- Assignment Variable $x_{i,j}$: A binary variable indicating whether item $i$ is assigned to bin $j$

$$\text{Minimize} \quad \sum_{j=1}^{n} y_j$$

$$\text{Subject to:}$$

$$\sum_{i=1}^{n} s_i x_{i,j} \le c y_j \quad \forall j = 1, \ldots, n$$

$$\sum_{j=1}^{n} x_{i,j} = 1 \qquad \forall i = 1, \ldots, n$$

$$x_{i,j}, y_j \in \{0,1\} \quad \forall i, j = 1, \ldots, n$$

---

## E.6. An Example of Generator

**Python Code for Bin Packing Generator**

```python
import gurobipy as gp
from gurobipy import GRB
import random

class Generator:
    def __init__(self, parameters=None, seed=None):
        self.problem_type = "binpacking"
        default_parameters = {
            "n_items": (3, 10),
            "weight_range": (1, 50),
            "bin_capacity": 100
        }
        if parameters is None:
            parameters = default_parameters
        for key, value in parameters.items():
            setattr(self, key, value)
        self.seed = seed
        if self.seed:
            random.seed(seed)

    def generate_instance(self):
        self.n_items = random.randint(*self.n_items)
        items = list(range(self.n_items))
        item_weights = {i: random.randint(*self.weight_range) for i in items}

        model = gp.Model("BinPacking")
        model.Params.OutputFlag = 0  # Suppress Gurobi output
        x = model.addVars(items, items, vtype=GRB.BINARY, name="x")
        y = model.addVars(items, vtype=GRB.BINARY, name="y")

        # Objective: Minimize the number of bins used
        model.setObjective(gp.quicksum(y[j] for j in items), GRB.MINIMIZE)
        for j in items:
            model.addConstr(
                gp.quicksum(item_weights[i] * x[i,j] for i in items) <= self.
    bin_capacity * y[j],
                name=f"Capacity_{j}"
            )
        for i in items:
            model.addConstr(
                gp.quicksum(x[i,j] for j in items) == 1,
                name=f"Assignment_{i}"
            )
        return model
```

## E.7. Augmentation Prompt

---

GENERATE AUGMENTATION PROBLEM PROMPT

---

```
AUGMENTATION_RULES = [
    # Semantic Enhancement
    "Rephrase the problem description while maintaining the same mathematical
    structure...",
    "Rewrite the problem using different expressions and terminology...",
    # Change the scenario
    "Transform the problem into a different application scenario while preserving
    the same structure...",
    "Conceive a variant on another scenario for the mathematical model...",
    # Numerical enhancement
    "Change the numerical parameters while maintaining the same problem structure
    ...",
    "Scale up or down the problem size by adjusting parameters proportionally...",
    # Problem variant generation
    "Generate a variant by adding/removing/modifying constraints...",
    "Create a variation by combining different types of constraints...",
    # Complicating the problem
    ## Variable Expansion
    "Increase the number of decision variables while maintaining similar structure
    ...",
    "Add bounds for adjustment variables...",
    ## Constraints Expansion
    "Add realistic constraints like capacity limitations, budget restrictions...",
    "Introduce cross-variable constraints between components...",
    ## Data Complexity
    "Convert parameters into tabular form with more complex data structures...",
    ## Problem Types
    "Generate non-linear problems by replacing linear relations...",
    "Combine with other problem types to generate hybrid problems...",
    # Multi-objective
    "Add new objective functions to generate multi-objective problems...",
    "Modify objective function to include additional terms...",
    # Problem symmetry
    "Generate variants by introducing symmetries...",
    "Generate dual problems while modifying parameters and constraints..."
]

AUGMENTATION_TEMPLATE = """Below is an optimization problem, please generate a new
    optimization problem by following the augmentation rule provided.

# Original Problem
The original optimization problem is as follows:
'''
{original_problem}
'''
# Augmentation Rule
{rule}
# Augmented Problem
Please construct a new optimization problem according to the above requirements and
    the provided question in the following format:

[Write your new problem here]

Note: The generated problem should maintain mathematical validity and practical
    feasibility.And just provide the problem description without any additional
    information.
"""
```

