# OpenReview forum: "OptMATH: A Scalable Bidirectional Data Synthesis Framework for Optimization Modeling"
_ICML.cc/2025/Conference — ICML 2025 poster_

### Official Review · Reviewer_PCa3 · 2025-03-10

**Overall Recommendation:** 3

**Summary:**

This paper proposes an automatic data synthesis framework for LLM optimization modeling. The method can control the problem complexity starting with some seed data. Then, the method obtains the natural language description using a backtranslation step. Experiments demonstrate the effectiveness of training various sizes of LLMs using the generated dataset.

**Claims And Evidence:**

The paper mentioned "This increased complexity, manifested through longer problem descriptions, poses greater challenges for LLMs." The author may want to investigate the relation between the modeling accuracy and the problem length. In my understanding, the most challenging part of LLM is to understand the scenarios of the problems, not necessarily the description length.

**Essential References Not Discussed:**

The author may want to cite the following works on LLM automatic modeling.

[1] OptiBench Meets ReSocratic: Measure and Improve LLMs for Optimization Modeling.

**Experimental Designs Or Analyses:**

1. Some experimental results are missing. I wonder whether the results of Chain-of-Experts/Optimus on the MAMO EasyLP/MAMO ComplexLP/OptMATH Bench are missing.
2. The author may want to conduct experiments on the harder dataset, either the IndustryOR or ComplexOR dataset.

**Methods And Evaluation Criteria:**

While using LLMs to analyze the LP files sounds interesting, this paper can be hard to generalize to large-scale instances. In practice, the LP files can be large, even more than 10M, which poses a great challenge for LLM to process such long input.

**Other Comments Or Suggestions:**

If the authors can address my concerns, I would like to increase my score.

**Other Strengths And Weaknesses:**

Strengths:

1. This paper proposes an interesting bidirectional data synthesis framework for optimization modeling.
2. The experiment results seem promising in LLM finetune.

Weaknesses:

1. The improvement in the Qwen models is significant. However,  to demonstrate the applicability to other LLMs, I suggest the authors provide experiment results on the Llama models.

**Questions For Authors:**

1. How about the data generation efficiency of the method?
2. How about the number of constraints and variables in the generated problems?

**Relation To Broader Scientific Literature:**

This paper is related to the LLM automatic formulation for mathematical optimization.

**Theoretical Claims:**

This paper does not contain any proof for theoretical claims.

---

> ### Author Rebuttal · Authors · 2025-04-01
>
> Thank you for the detailed feedback. We address the specific questions:
>
> **Regarding Claims And Evidence:**
>
> - **Problem Length vs. Complexity:** We fully agree that scenario understanding is crucial for LLMs. However, more complex scenarios naturally require longer, more detailed NL descriptions (e.g., ROADEF 2012 [1]), challenging both LLM abstract reasoning and long-context processing. Thus, length reflects certain scenario complexity. Our paper's results on MAMO (Fig 3, Table 1) show a correlation between scale/length and reduced accuracy.
> - **Scenario/Problem Type Coverage:** Our seed dataset features **over 50 expert-curated problem classes**, which we believe provides substantial and representative coverage. As detailed in Appendix A.2, each class is grounded in **referenced literature and reflects practical application scenarios**. While methods like the evolutionary approach in Li et al. [2] can generate synthetic variations, they often start from a limited set of initial classes (reportedly 8 in that case), whereas our foundation of **50+ manually curated classes is significantly more extensive**. Moreover, such synthetic generation typically recombines existing elements rather than creating fundamentally new problem types. We are confident in the breadth of our 50+ classes; further expansion could effectively build upon this rich seed set using augmentation or evolution.
>
> **Regarding Methods And Evaluation Criteria:**
>
> - **Scalability for Massive Instances:** Our current pipeline already handles considerable scales effectively (up to ~25k characters, Fig 2). However, for the ultra-large instances (>10MB) you mentioned, direct LLM processing is indeed problematic due to context limits. To address this, our framework can adapt using a Model-Data Separation Format (similar to OptiBench [3]):
>     - **Format:** Instance = (NL Description + Structured Data File). NL has scenario/data needs; Data File has numerics.
>     - **OptMATH Adaptation:** Generate (NL + Data File) pairs. Train AutoFormulator to generate code reading the Data File, validating via OV or bipartite graph check. The (MF, NL, PD) interpretation shifts: NL includes scenario + data ref; PD is the final solver-ready format. This is a promising direction for ultra-large scales.
>
> **Regarding Experimental Designs Or Analyses:**
>
> - **Missing Results & Harder Datasets:** We added the requested Optimus results on MAMO/OptMATH Bench and other benchmarks. We also added IndustryOR and OptiBench results. See Table on the response to **Reviewer mCmd.**
> - **Regarding Chain-of-Experts and ComplexOR:** We do not show the results for Chain-of-Experts (CoEs) [5] and ComplexOR. The input for the CoEs requires each instance to include a structured code template and detailed descriptions of its parameters. Additionally, the ComplexOR dataset format is not suitable for end-to-end modeling tests, as its data format for each problem includes a natural language description without numerics.
>
> **Regarding Essential References Not Discussed:**
>
> - We agree [4] is pertinent and will add/discuss it in revised Sec 1. While both explore reverse synthesis, OptMATH uniquely emphasizes:
>     - **Generator-based Scalability:** Abstract MF + parameterized generators enable large-scale, diverse PD creation (Sec 3, Alg 1).
>     - **Rigorous Semantic Validation:** Strict Optimal Value (OV) equivalence check (Sec 4.3) ensures high NL-PD correctness (99.6% manual accuracy), beyond just code executability.
>
> **Regarding Weaknesses:**
>
> - **Applicability to Llama:** We fine-tuned Llama 3.1 8B on OptMATH-Train, showing applicability. Table of the response to Reviewer mCmd shows significant improvements over baseline Llama. Performances of checkpoints are uploaded in [6] named llama_checkpoints_perf.
>
> **Regarding Questions For Authors:**
>
> - **Data Generation Efficiency:** Uses feedback-driven tuning (Alg 1) for controllable quality. PD generation success ~50% (see [6], rate_after_feedback_tuning figure). Rejection sampling (OV matching validation) acceptance is 62.14% (Fig 14: T=1 efficient). 200k valid (NL, PD) samples cost ~$1914 (13.35B tokens, DeepSeek-V3, promotional period), much cheaper than manual creation.
> - **Number of Constraints/Variables:** OptMATH-Train covers wide range: up to ~2500 vars / ~1800 cons, including complex instances. Please refer to optmath_train_under500, optmath_train_linear, optmath_train_log, benchmarks_distribution, benchmarks_under100, and benchmarks_box_plot figures in [6] for a more detailed description.
>
> ---
>
> **References:**
>
> [1] ROADEF Challenge 2012 Subject.
>
> [2] Towards foundation models for mixed integer linear programming.
>
> [3] OptiBench: A Large Language Model Benchmark for Optimization Problem Understanding and Formulation.
>
> [4] OptiBench Meets ReSocratic: Measure and Improve LLMs for Optimization Modeling.
>
> [5] Chain-of-Experts: When LLMs meet complex operations research problems.
>
> [6] https://anonymous.4open.science/r/OptMATH-Rebuttal-8F5E

---

### Official Review · Reviewer_LZQX · 2025-03-14

**Overall Recommendation:** 3

**Summary:**

This paper proposes OptMATH, a method for data generation in the field of optimization modeling, which primarily combines the "Back Translation" technique from previous work with the "rejection sampling" method. Strictly speaking, it falls under the categories of data augmentation and data annotation within data generation.

Additionally, it introduce a benchmark that is more comprehensive compared to others in the field, capable of varying difficulty levels and providing stricter scoring, thereby better assessing a model's optimization modeling capabilities.

Overall, a substantial amount of experimentation has been conducted, covering all necessary aspects, and the experimental results are promising, even surpassing those of GPT-4.

**Claims And Evidence:**

Please refer to Questions For Authors

**Essential References Not Discussed:**

No

**Experimental Designs Or Analyses:**

The submission contains some claims that are not fully supported by clear and convincing evidence. Specific problematic claims will be addressed in the "Other Strengths And Weaknesses" section.

**Methods And Evaluation Criteria:**

The methods and metrics make sense.

**Other Comments Or Suggestions:**

No

**Other Strengths And Weaknesses:**

Advantages:
1. The paper introduces OptMATH, a method for data generation in optimization modeling integrating "Back Translation" with "rejection sampling", contributing to solving the data shortage in field of optimization modeling.
2. Besides, the introduction of OptMATH-Bench provides a more robust assessment of a model's optimization capabilities compared to existing benchmarks.
3. The extensive experimentation conducted covers many necessary aspects of the study, yielding promising results that even surpass those of GPT-4.

Disadvantages:
1. The article lacks transparency in explaining how the rejected data was refined to establish the proposed benchmark, leaving a gap in understanding the curation process.
2. The paper fails to clearly articulate where the controllable difficulty aspect of the research is manifested. While mentioning the definition of difficulty levels and data generation through a generator, the specifics of training, generation, and validation of the produced data remain undisclosed. This lack of detail raises concerns about the robustness and reliability of the generated seed dataset.
3. The utilization of data from challenging benchmarks as seed data, without a comprehensive explanation or validation process, raises questions about the fairness and reasonableness of the experimental setup. Testing on different benchmarks, even if not identical, could potentially introduce biases or skew results.
4. Insufficient elaboration on the methodology employed to create OptMATH-Bench limits the clarity on its distinctiveness from the training data generation process. Without a clear distinction in sources, distributions, and methodology, achieving superior performance on OptMATH-Bench may primarily reflect strong in-domain capabilities rather than a comprehensive model evaluation.

**Questions For Authors:**

1. They did not provide a detailed explanation of how the rejected data was further curated to create the proposed benchmark.
2. It is not clearly stated where the controllable difficulty of this work is reflected. They simply mentioned defining difficulty levels and using a generator to produce data? My understanding is that they classify the difficulty of existing data, then train a generator to produce corresponding PD, MF, and OV based on difficulty requirements, which are then fed into the data generation pipeline. However, the specific training and generation details were not further disclosed. Additionally, I assume these data serve as the seed dataset. Was there further validation to ensure the reasonableness and correctness of the generated seed data?
3. Part of the data source comes from challenging benchmarks. Generating data from other benchmarks and then testing on different benchmarks, even if not the same one, raises questions: Is this setup reasonable? Is it fair?
4. The article does not provide further details on the method for creating OptMATH-Bench. My understanding is that if the methods for generating training data and the benchmark are largely similar, and their sources and distributions are consistent, then good performance on OptMATH-Bench may only indicate strong in-domain capabilities. Outperforming other models on this benchmark might not be entirely fair.

**Relation To Broader Scientific Literature:**

How are the key contributions of the paper related to the broader scientific literature? Be specific in terms of prior related findings/results/ideas/etc.
1. This article falls within the domains of data augmentation and data annotation in data generation, specifically under the broader category of synthetic data generation.
2. The techniques it employs, namely "back translation" and "rejection sampling," are not uncommon in the field of synthetic data generation.

**Theoretical Claims:**

This is a work of data synthesis, so there is no proof in the article, and many formulas are just explanations or definitions of properties.

---

> ### Author Rebuttal · Authors · 2025-04-01
>
> Thank you for the detailed feedback. We address the specific questions raised:
>
> **1. Regarding Questions 1 & 4: OptMATH-Bench Curation, Distinctiveness, and In-Domain Evaluation**
>
> We clarify OptMATH-Bench's curation and distinction from OptMATH-Train to address concerns about evaluating only in-domain capabilities:
>
> - **Dual Curation Pathways:** OptMATH-Bench was created via two distinct routes. **Pathway 1** started with instances **rejected** by our AutoFormulator (failed OV check), indicating initial difficulty. An "LLM-Committee" (inspired by [1], using diverse powerful models like GPT-4, Claude, Gemini, DeepSeek) then filtered these: **(PD, NL) pairs** were retained only if **at least one and at most two** committee members successfully formulated them (passed OV check). This isolated well-posed but non-trivial modeling challenges. Crucially, **human OR experts** subsequently **validated the correctness** of these selected pairs and further refined them based on relevance and clarity. **Pathway 2** involved **experts directly curating challenging problems from external OR literature** (journals, textbooks), ensuring methodological/source independence and including known hard problem types (e.g., NLP, SOCP - see Figure 4).
> - **Addressing "In-Domain" Concern:** This dual approach ensures distinction. Pathway 2 uses external sources/methods. Pathway 1 involves significant expert validation. Even if underlying PD distributions overlap, the **distribution of (PD, NL) pairs** in OptMATH-Bench is fundamentally different due to curation. The LLM-Committee filtering specifically ensures the benchmark represents the **hard tail of this paired distribution** relative to strong LLM capabilities. The superior performance of our finetuned model on newly added IndustryOR and OptiBench benchmarks (see Table of the response to **Reviewer mCmd**) further contradicts a purely in-domain evaluation and shows generalization ability.
>
> *Proposed Revision:* We will revise Section 4.3/6.1 to detail these distinct curation pathways, emphasizing expert roles, external sourcing, the LLM-committee filtering targeting the (PD, NL) hard tail, and problem diversity.
>
> **2. Regarding Question 2: Controllable Difficulty Mechanism and Seed Data Validation**
>
> We clarify the difficulty control and validation, addressing a misunderstanding about generator training:
>
> - **Generator Usage Clarification:** We must correct a misunderstanding: we do **not** *train* generators. We use **pre-defined, parameterized code generators** implementing standard MFs from OR literature (e.g., Bin Packing [2], Appendix E.6). Our novelty is controlling their *input parameters*.
> - **Feedback-Driven Parameter Tuning (Alg. 1):** Difficulty is controlled via an iterative LLM feedback loop (inspired by [3]). The LLM suggests/refines generator parameters based on evaluation feedback (complexity score S(PD), solve time, feasibility) from generated instance batches, steering output towards targets.
> - **Further Validation:** AutoFormulator training uses **curriculum learning**. Seed MFs/generators are expert-curated from literature (Appendix A.2) **to ensure the reasonableness and correctness**. PDs are validated for feasibility/solvability (Alg. 1) and OV equivalence (Alg. 2).
>
> *Proposed Revision:* We will revise Sections 3, 5.2, Appendix A/B to detail the LLM-driven *parameter tuning* (not generator training), curriculum learning, seed curation, and validation.
>
> **3. Regarding Question 3: Use of Benchmarks for Seed Inspiration and Experimental Fairness**
>
> We clarify the use of solving benchmarks like MIPLIB and address fairness:
>
> - **Solving Benchmarks as Inspiration Only:** These were used *solely* to identify **representative OR problem classes** (e.g., TSP, Job Shop). Their specific solver-focused PD (lacking NL) were **not** reused.
> - **Independent Generator Development:** We built **new parameterized generators from scratch** based on MFs from classic OR **literature** for each identified class (e.g., Job Shop [4], Appendix A.2).
> - **Distinction Justifies Fairness:** The clear separation – using solver benchmarks only for class inspiration, building new generators from literature, not reusing instances, and testing on different types of benchmarks (Optimization Modeling vs. Solving) – ensures our setup is reasonable and fair. We argue that **this strikingly contradicts hacking the modeling benchmarks**. We do not use any benchmarks directly.
>
> *Proposed Revision:* We will revise Section 3 and Appendix A.2 to explicitly state this distinction and the generator development process based on literature.
>
> ---
>
> **References:**
>
> [1] Auto-Arena: Automating LLM Evaluations with Agent Peer Battles and Committee Discussions.
>
> [2] Analysis and design of algorithms in combinatorial optimization.
>
> [3] Large language models as optimizers.
>
> [4] The shifting bottleneck procedure for job shop scheduling.

---

### Official Review · Reviewer_mCmd · 2025-03-14

**Overall Recommendation:** 3

**Summary:**

The paper proposes a framework named OptMATH for synthesizing high-quality datasets aimed at optimization modeling from natural language descriptions. This framework addresses the scarcity of optimization datasets by generating problem data through mathematical formulations and back-translation into natural language descriptions. The framework includes a rigorous quality control process involving forward modeling and rejection sampling to ensure mathematical consistency. Extensive experiments demonstrate that models trained on the OptMATH dataset outperform existing benchmarks, showcasing the framework's effectiveness and scalability.

**Claims And Evidence:**

Experimental results have shown the proposed method can achieve strong results compared with previous studies.

**Essential References Not Discussed:**

N/A

**Experimental Designs Or Analyses:**

Yes

**Methods And Evaluation Criteria:**

Yes, the proposed method mainly addresses the data scarcity issue in optimization modeling.

**Other Comments Or Suggestions:**

N/A

**Other Strengths And Weaknesses:**

### Strengths
1. The proposed bidirectional data synthesis framework is innovative and provides a systematic solution to the issue of data scarcity in optimization modeling.
2. The use of rejection sampling ensures high-quality data generation, with the framework demonstrating a remarkable 99.6% accuracy in maintaining mathematical consistency.
3. The framework is versatile, covering over 10 real-world applications with various optimization problems such as LP, MILP, IP, NLP, and SOCP.
4. Comprehensive experiments validate the framework's effectiveness, with models trained on OptMATH achieving superior performance on multiple established modeling benchmarks, including NL4Opt and MAMO.

### Weaknesses
1. While the authors highlight limitations in previous prompting-based methods, the paper lacks clear articulation of how OptMATH specifically overcomes these issues. In the 'Our Contributions' section, consider adding concise explanations that directly contrast OptMATH's advancements with those of prior work, providing readers with a clearer understanding of its unique advantages.
2. The experiments primarily focus on NL4Opt and MAMO benchmarks. Evaluating the framework on a wider range of standard datasets would be better.
3. To better illustrate the performance gains achieved by OptMATH, consider including results for Qwen2.5-7B and 32B in Table 1. This would provide a more comprehensive comparative analysis.
4. The selection of seed data is a critical aspect of OptMATH. The paper should dedicate more space to detailing the process of seed data selection, including the criteria and methodologies employed.

**Questions For Authors:**

N/A

**Relation To Broader Scientific Literature:**

I cannot provide an accurate assessment towards the broader scientific literature, as I am not sufficiently familiar with the target literature.

**Theoretical Claims:**

I did not find any flaws currently.

---

> ### Author Rebuttal · Authors · 2025-04-01
>
> Thank you for your valuable feedback. We appreciate the suggestions for improving the clarity and scope of our work. We address each point below:
>
> **Regarding Weakness 1:** We will revise the **'Our Contributions' section** to more explicitly contrast OptMATH with prior prompting-based methods. Key differentiators that will be emphasized include:
>
> - **Specialized & Efficient Models:** Fine-tuning on OptMATH-Train yields specialized AutoFormulator models demonstrating superior performance. Additionally, complex, multi-step prompting pipelines (e.g., multi-turn interaction) can sometimes confuse models on simpler tasks, potentially degrading performance relative to complex ones (see Table, OptiMUS on MAMO  EasyLP). Furthermore, fine-tuning yields models with low inference costs suitable for large-scale deployment, unlike prompt-based methods requiring continuous, expensive calls to powerful foundation model APIs.
> - **Beyond Prompting:** Unlike methods relying solely on base LLM capabilities via prompting, OptMATH focuses on fine-tuning models using a large-scale, high-quality dataset synthesized specifically for optimization modeling. This fine-tuning approach **is complementary to prompt engineering techniques**, offering a path to enhance foundational model capabilities rather than just leveraging existing ones.
> - **Scalable High-Quality Data Generation:** Our bidirectional framework systematically generates vast amounts of verified (NL, MF, PD) triplets, addressing the data scarcity that limits fine-tuning approaches.
> - **Rigorous Semantic Validation:** We employ Optimal Value (OV) based rejection sampling, ensuring semantic consistency between the NL description and the problem data, which is more rigorous than checks for mere code executability often used implicitly by prompting methods.
> - **Controllable Complexity & Diversity:** The framework allows generating problem data with targeted difficulty via feedback loops and incorporates diverse problem types.
>
> **Regarding Weaknesses 2&3:** We now include results on the IndustryOR [1] and OptiBench [2] benchmarks. Our Qwen2.5-32B model, finetuned on the OptMATH-Train dataset, demonstrates consistently superior performance, achieving results comparable to GPT-4 and DeepSeek-V3. Results for OptiMUS, Llama, and ORLM on these benchmarks are also presented. Furthermore, to illustrate the impact of finetuning, results for the baseline Llama and Qwen models (before finetuning) have been added. This updated table provides a comprehensive comparative analysis. Due to time constraints, the reproduced ORLM results were neither tuned nor run multiple times, which explains the drop in performance.
>
> | Models | NL4OPT | MAMO  EasyLP | MAMO  ComplexLP | OptMATH-Bench | IndustryOR | OptiBench |
> | --- | --- | --- | --- | --- | --- | --- |
> | GPT-3.5-turbo | 78.0%  | 79.3%  | 33.2%  | 15.0% | 21.0% | 58.1% |
> | GPT-4 | 89.0%  | 87.3%  | 49.3%  | 16.6% | 33.3% | 68.6% |
> | DeepSeek-V3-1226 | 95.9%  | 88.3%  | 51.1%  | 32.6% | 37.0% | 71.6% |
> | OptiMUS base on GPT-4o (2024-05-13) | 78.8% | 77.0% | 43.6% | 20.2% | 31.0% | 45.8% |
> | LLama3.1_8B (pass@1) | 0% | 0.2% | 0% | 0% | 0% | 0% |
> | OptMATH_LLama3.1_8B (pass@1) | 55.5% | 73.9% | 40.8% | 24.4% | 18% | 55.5% |
> | OptMATH_LLama3.1_8B (pass@8) | 97.6% | 94.2% | 71.6% | 51.6% | 37% | 66.6% |
> | Qwen2.5_7B (pass@1) | 86.9% | 83.6% | 21.8% | 1.6% | 10% | 36.2% |
> | OptMATH_Qwen2.5_7B (pass@1) | 94.7%  | 86.5%  | 51.2%  | 24.4% | 20% | 57.9% |
> | OptMATH_Qwen2.5_7B (pass@8) | 98.4% | 94.5% | 72.5% | 56.0% | 38.0% | 68.1% |
> | Qwen2.5_32B (pass@1) | 92.7% | 82.2% | 44.6% | 9.3% | 16.0% | 47.6% |
> | **OptMATH_Qwen2.5_32B (pass@1)** | **95.9%** | **89.9%** | **54.1%** | **34.7%** | **31.0%** | **66.1%** |
> | OptMATH_Qwen2.5_32B (pass@8) | 97.9% | 93.9% | 75.4% | 67.4% | 47.0% | 76.8% |
> | ORLM-LLaMA-3-8B (reported) | 85.7% | 82.3% | 37.4% | * | 38.0% | * |
> | ORLM-LLaMA-3-8B (reproduced) | 84.5% | 74.9% | 34.1% | 2.6% | 24.0% | 51.1% |
>
> **Regarding Weaknesses 4:** We agree this is important. The comprehensive methodology, criteria, and structured organization for our seed data generation—including how we utilize benchmark problem structures (e.g., from MIPLIB) to create validated, parameterized instance generators and associated metadata—are detailed in **Appendix A.2 (Seed Classes)**. Recognizing the value of highlighting this in the main text, we will revise **Section 3** to include a concise summary of this systematic process and clearly reference Appendix A.2 for the full description.
>
> ---
>
> References:
>
> [1] ORLM: Training large language models for optimization modeling.
>
> [2] OptiBench meets ReSocratic: Measure and improve LLMs for optimization modeling.

---

### Official Review · Reviewer_WcAc · 2025-03-18

**Overall Recommendation:** 3

**Summary:**

The paper presents OptMATH, a scalable bidirectional data synthesis framework designed to address the challenge of data scarcity in optimization modeling. It automatically generates high-quality optimization problem data with controllable complexity, starting from curated seed data with mathematical formulations. The framework employs a backtranslation step to obtain natural language descriptions and uses forward modeling with rejection sampling to verify the correspondence between the descriptions and problem data. The accepted pairs form the OptMATH training dataset, while rejected pairs are filtered to create a challenging benchmark. Extensive experiments demonstrate that models trained on OptMATH achieve superior results on multiple modeling benchmarks, validating the framework's effectiveness and scalability.

**Claims And Evidence:**

In general, the claims made in the submission are supported by clear and convincing evidence.

**Essential References Not Discussed:**

OptiBench Meets ReSocratic: Measure and Improve LLMs for Optimization Modeling. ICLR 2025

**Experimental Designs Or Analyses:**

See weaknesses below

**Methods And Evaluation Criteria:**

The proposed methods and evaluation criteria are well-suited for the problem.

**Other Comments Or Suggestions:**

No

**Other Strengths And Weaknesses:**

While the OptMATH framework presents a significant advancement in generating high-quality optimization modeling datasets, there are several potential weaknesses and areas for improvement:

1. Complexity of Natural Language Descriptions: The paper acknowledges that the complexity of natural language descriptions can vary widely. While the framework aims to generate high-quality descriptions, there may still be instances where the descriptions are overly complex or ambiguous, making it difficult for models to accurately translate them into mathematical formulations.

2. Generalization to New Domains: Although the framework covers a wide range of optimization problems, its ability to generalize to entirely new domains or problem types not covered in the seed data is uncertain. The framework relies on curated seed data, which may limit its adaptability to novel optimization scenarios.

3. Computational Resources: The bidirectional synthesis process, including feedback-driven problem data generation and rejection sampling, is computationally intensive. This may limit the scalability of the framework, especially for very large datasets or extremely complex optimization problems.

4. Optimization Problem Diversity: While the framework generates a diverse set of optimization problems, there is a risk that the generated problems may not fully capture the diversity of real-world optimization challenges. The reliance on existing benchmarks and expert-curated seed data might introduce biases or limit the range of problem types generated.

5. Evaluation Metrics: The paper uses accuracy (pass@1) as the primary evaluation metric, which measures whether the optimal value obtained by the generated code matches the ground truth. While this is a relevant metric, it may not fully capture the quality of the generated mathematical formulations or the reasoning capabilities of the models. Additional metrics, such as the diversity of generated problems or the robustness of the models to variations in problem descriptions, could provide a more comprehensive evaluation.

6. Human-in-the-Loop: The framework involves a significant amount of human effort in curating seed data, designing prompt templates, and validating the generated datasets. This reliance on human expertise may limit the framework's ability to be fully automated and could introduce human biases.

7. Solution-Based Validation: The rejection sampling mechanism relies on comparing the optimal values of the original and generated problem instances. While this approach ensures a high degree of semantic equivalence, it may not guarantee the exact equivalence of the mathematical formulations. Further research is needed to develop more sophisticated validation techniques that can ensure the precise correspondence between natural language descriptions and mathematical formulations.

8. Model Size and Data Scaling: The paper demonstrates that larger models generally achieve better performance, but the relative gains from fine-tuning diminish as model size increases. This suggests that there may be diminishing returns in scaling up the model size, and more efficient training strategies or model architectures might be needed to achieve better performance on complex optimization tasks.

Overall, while the OptMATH framework represents a significant step forward in generating high-quality optimization modeling datasets, addressing these weaknesses could further enhance its effectiveness and applicability in real-world scenarios.

**Questions For Authors:**

No

**Relation To Broader Scientific Literature:**

Optimization.

**Theoretical Claims:**

This paper did not provide any proofs for theoretical claims.

---

> ### Author Rebuttal · Authors · 2025-04-01
>
> Thank you for your thoughtful review on the potential areas for improvement. We appreciate the opportunity to address these points:
>
> **Essential References Not Discussed:** See “Essential References Not Discussed” section of response to Reviewer PCa3
>
> **Regarding Weakness 1:** Our framework directly addresses this via the **rejection sampling mechanism detailed in Section 4.3 and Algorithm 2**. Each generated NL description is translated back into problem data (PD') using our AutoFormulator. We then rigorously validate this translation by comparing the optimal objective value (OV') obtained from solving PD' against the optimal value (OV) from the original problem data (PD). Only pairs where OV' equals OV are accepted into OptMATH-Train. Consequently, this process **can process NL descriptions that seem to be complex or ambiguous** accurately, ensuring that all included data points feature corresponding NL and PD that are **demonstrably modelable by an LLM**.
>
> **Regarding Weakness 2:** Please refer to the "Addressing 'In-Domain' Concern" and "Distinction Justifies Fairness" sections of the response to Reviewer LZQX and the table in the response to Reviewer mCmd, where we add new benchmarks. These arguments clearly show that the **benchmarks, even OptMATH-Bench, originating from the same PD distribution, do not overlap with OptMATH-Train**. These results show the models finetuned on OptMATH-Train **generalize to entirely new domains or problem types beyond the curated seed data.**
>
> **Regarding Weakness 3:** Please refer to the "Scalability for Massive Instances" a "Data Generation Efficiency" sections of the response to Reviewer PCa3. Our pipeline is efficient and can **easily adapt the data-model separation paradigm** to process extremely complex optimization problems.
>
> **Regarding Weakness 4:** Please refer to the "Scenario/Problem Type Coverage" section of the response to Reviewer PCa3.
>
> **Regarding Weakness 5:** In **Appendix A.2 and A.3, we have given a detailed description of the diversity** of seed data and OptMATH-Train. For additional metrics, please refer to the table in the response to Reviewer mCmd. We evaluate the performance of finetuned models on IndustryOR and OptiBench. We also reported the results using pass@8; the higher performance at pass@8 compared to pass@1 indicates a **high upper bound on the modeling capability** of the finetuned models, signifying that models trained on OptMATH-Train attain a **high ceiling for their modeling skills**.
>
> **Regarding Weakness 6:** We agree that this is a critical aspect and will revise the paper to elaborate on our methodology. To clarify, our **process is largely automated**: we begin with core problem structures inspired by benchmarks like MIPLIB (e.g., TSP, JSS variants) and **utilize parameterized instance generators** designed around these structures. For instance, our Job Shop Scheduling generator accepts parameters like job/machine counts and operation details to automatically create problem data (PD). Crucially, these **generated PD are validated for solvability (e.g., using Gurobi feasibility checks) before being included in our seed set** and subsequently translated into natural language by LLMs. To further increase automation and reduce human biases, we can easily adapt the methods in addressing **weakness 4** and **reduce the amount of seed data to merely 8 generators**.
>
> **Regarding Weakness 7:** Our empirical validation is crucial here: as reported in Section 4.3, manual checks confirmed that our OV-based rejection sampling **achieves a 99.6% accuracy rate** in capturing the correct problem semantics and ensuring practical equivalence. We find this level of accuracy to be **highly effective and practically sufficient** for ensuring dataset quality for LLM training. Alternatively, checking graph isomorphism for LP problem structures, inspired by related work [1], could be used for stricter MF equivalence; however, we **opted for the OV check due to its operational simplicity and broad applicability** across problem types within our framework, ensuring a practical and scalable validation approach.
>
> **Regarding Weakness 8:** To proactively enhance scalability and mitigate this effect, our framework incorporates specific strategies aimed at maximizing data diversity within OptMATH-Train. As detailed in **Section 5.1, we employ extensive data augmentation techniques** to generate more varied and non-standard problem instances. Additionally, during the forward modeling phase [**Section 4.2**], **we utilize diverse Chain-of-Thought (CoT) prompting strategies** to capture multiple valid reasoning paths and formulation variants. We believe that enriching the training data with such diversity **helps elicit stronger performance improvements even on larger models, thereby counteracting the diminishing returns trend**.
>
> ---
>
> **References**
>
> [1] OptiBench: A Large Language Model Benchmark for Optimization Problem Understanding and Formulation.

---

> > ### Comment · Reviewer_WcAc · 2025-04-04
> >
> > Thanks for the clarifications. I will keep my rating as weak acceptance for this paper.

---

### Decision · Program_Chairs · 2025-05-01

**Decision:**

Accept (poster)

**Comment:**

The paper presents a bidirectional data synthesis framework, termed  OptMATH, for constructing datasets on optimization modeling tasks. It starts from curated seed data in mathematical formulations, then employs a backtranslation to obtain natural language descriptions. Further, OptMATH uses forward modeling with rejection sampling to verify the correspondence between the descriptions and problem data. The accepted pairs form the training dataset, while the rejected pairs are filtered to create a challenging benchmark. Experiment results show that models trained on OptMATH achieve better performance on various benchmarks.

As the reviewers point out, the proposed method contributes to solving the data shortage challenge in the field of optimization modeling. The idea of bidirectional data synthesis is interesting. The experiment results show that the synthesized data is promising in improving the modeling ability of LLM. All reviewers give positive scores, though no strong support is provided. I recommend acceptance.